# KGR-SKATER: Spatially clustered kernel graph regression for counting processes

Jeffrey Wu[¤]*, Gareth W. Peters[¤]*, Alex Franks[¤]*

Department of Statistics & Applied Probability, UCSB, Santa Barbara, California, United States of America

¤ Current address: Santa Barbara, California, United States of America
* jeffreywu@pstat.ucsb.edu (JW); garethpeters@pstat.ucsb.edu (GWP); afranks@pstat.ucsb.edu (AF)

## Abstract

This paper proposes a procedure for fitting a spatiotemporal model with an interpretable and parsimonious dependence structure to high-dimensional non-Gaussian data. A graph is estimated to represent spatial dependence, and a locally periodic kernel is estimated to represent temporal dependence. These two components are then combined via a Kronecker product, producing a separable spatiotemporal covariance matrix that can account for multiple relevant variables and their dependencies at different time scales. Spatial clustering is used to reduce the dimensionality and estimation is carried out via the integrated nested Laplace approximation. The proposed model, which relies on all of these preceding steps, is introduced along with some alternative spatiotemporal reference models for comparison. The utility of this novel multi-step procedure is demonstrated by modeling monthly time series of respiratory-related deaths across California. Social deprivation indices are used to learn a graph structure, and surrogate variables constructed from exposure adjusted measures of air quality are used to learn time series regression relationships encoded by a kernel. The modeling results indicate that KGR-SKATER models fit the data in and out of sample as well as the reference models and have better coverage properties. A synthetic case study is also presented to demonstrate how the proposed procedure makes better forecasts than the reference model in settings where time series exhibit less stationary amplitudes and periods. Data and code available at: https://doi.org/10.5061/dryad.j3tx95xtt.

## 1 Introduction

Modeling point processes is particularly challenging when dealing with heterogeneous and high-dimensional spatiotemporal dependence structures. This paper proposes a novel framework for modeling complex spatiotemporal processes in a way that reduces dimensionality, accounts for spatiotemporal autocorrelation, and accommodates exogenous and endogenous time series regression structures. The approach shows particular promise for complex spatiotemporal data with evolving

**Data availability statement:** The datasets and code used for this project are publicly available at https://datadryad.org/dataset/doi:10.5061/dryad.j3tx95xtt or https://github.com/jeffwu25/KGR-SKATER.

**Funding:** The author(s) received no specific funding for this work.

**Competing interests:** The authors have declared that no competing interests exist.

dependencies, offering a practical solution to the challenging problem of modeling heterogeneous covariance structures in space and time. This approach is motivated by an important real-world application aimed at characterizing how spatial patterns in socioeconomic indicators and temporal patterns in air pollutant levels drive respiratory-related mortality in California.

The field of point process analysis has evolved significantly since the early works of [1–3]. Key contributions to the field include the Besag-York-Mollie (BYM) model from [4], adapted from Bayesian image restoration, spatial generalized linear mixed models (GLMM) with spatial random effects [5], and Log Gaussian Cox processes (LGCP) from [6] for modeling inhomogeneous and dependent spatiotemporal data.

Modeling complex spatiotemporal dependence structures is especially challenging and computationally expensive for high-dimensional non-Gaussian data. Non-sparse covariance matrices are difficult or even intractable to estimate in high-dimensional cases where there are many autocorrelations to account for. Furthermore, flexible response distributions are necessary to account for various over and under dispersion patterns. A latent Gaussian model with a nontrivial dependence structure specification will be computationally expensive to estimate without some assumptions and/or approximations. Some recent techniques to facilitate estimation include the integration of adaptive radial basis functions in spatial GLMMs [7], the use of Integrated Nested Laplace Approximation (INLA) [8] for efficient Bayesian inference, and the combination of kernel regression and graph signal processing for improved prediction models [9–11].

This paper presents a novel procedure that combines the following useful techniques: (1) Spatial clustering with the Spatial 'K'luster Analysis by Tree Edge Removal (SKATER) algorithm [12] which reduces dimensionality in a way that has interpretative value; (2) graph estimation with the HUGE package [13] which allows one to estimate a graph on a relevant exogenous or endogenous variable instead of assuming a known graph/neighborhood structure; (3) kernel graph regression (KGR) which produces a separable, interpretable spatiotemporal covariance that incorporates both spatial and temporal dependencies in a way that is not as computationally expensive to estimate (see [14–16]); and (4) approximate Bayesian inference with INLA which provides an efficient alternative to MCMC that is about as accurate for estimating latent Gaussian models. The combination of these steps into a single framework has not been formalized before; therefore, it will be referred to henceforth as KGR-SKATER.

Although each component of the proposed procedure draws on existing methodologies, e.g., spatial clustering (SKATER), graphical model estimation, kernel-based Gaussian process regression, and approximate Bayesian inference via INLA, the novelty of the KGR-SKATER framework lies in the integration of these components into a unified modeling pipeline for high-dimensional spatiotemporal count data. In particular, the proposed framework introduces three methodological contributions. First, it combines spatial clustering with data-driven graph learning so that spatial dependence is inferred from covariates rather than imposed through geographic adjacency alone. Second, it introduces a graph-filtered Gaussian process prior constructed via the Kronecker product of a filtered graph Laplacian and a temporal kernel

Gram matrix, producing a parsimonious, interpretable, and separable spatiotemporal covariance structure. Third, the framework demonstrates how these components can be estimated efficiently within the INLA framework, allowing flexible kernel-based spatiotemporal models to be fit without relying on computationally expensive MCMC methods.

This procedure allows for thoughtful estimation of a flexible dependence structure that can account for multiple relevant variables and their dependencies at different time scales. The spatial dependence is data-driven and is learned from one subspace of variables using the 'skater' and 'huge' packages. The temporal dependence is attributed to cross-temporal structure and is learned in a nonstationary fashion on another subspace of variables with kernel functions. These two different ways of measuring dependence are combined in a computationally efficient manner via a Kronecker product. KGR-SKATER incorporates all these steps and is still relatively straightforward to implement. However, there are many choices that must be considered carefully within each step, allowing a model to be customized for a given application study, as explained later in this paper.

Recent developments in spatiotemporal modeling have explored kernel-based approaches that exploit sparse precision matrices constructed from local neighborhood interactions. For example, the stochastic local interaction (SLI) models proposed by [17,18] construct sparse precision matrices for space-time interpolation using local interaction energies. While these models also leverage graph-based representations, they focus on constructing sparse precision structures directly, whereas the KGR-SKATER framework emphasizes learning a spatial graph from covariates and combining it with kernel-based temporal dependence through a Kronecker product covariance structure.

The KGR-SKATER methodology is applied to model respiratory-related mortality across California over six years, considering socioeconomic status measured by Social Deprivation Index (SDI) as a spatial component and air quality variations as a temporal component. It is well established that these variables have associations with respiratory-related disorders and death, see [19–22]. But more importantly, as shown later in Fig 2 in Section 7.1, SDI has significant spatial variation but varies slowly in time. In contrast, air pollution measurements have significant temporal variation but do not vary much across California. This enables KGR-SKATER to distinguish temporal nonstationarity driven by air quality fluctuations from spatial nonstationarity linked to socioeconomic conditions, facilitating a cleaner decomposition of spatial and temporal dependence structures in the model.

The remainder of the paper is structured as follows: in Section 2, notation is introduced and the steps to model spatial dependence are described. In Section 3, the proposed model is introduced, along with a description of kernels for modeling temporal dependence. In Section 4, a few benchmark reference models are presented. In Section 5, a brief review of INLA and a description of how it is used to expedite the estimation of the proposed model are provided. In Section 6, an outline of the simulations conducted to validate the KGR-SKATER framework is provided, in addition to some practical implementation considerations. In Section 7, the proposed modeling procedure is applied in full to model the number of respiratory-related deaths from 2015 to 2019 in California at the monthly county level. In Section 8, a discussion of the main results of the paper and an outlook for future research are laid out. Finally, in the supporting information appendices, some additional experiments and findings are included that may be of interest to those who want to understand the finer details of the KGR-SKATER framework.

## 2  Components comprising KGR-SKATER approach

The KGR-SKATER framework proceeds through six main stages that jointly construct a parsimonious spatiotemporal dependence structure.

1. Spatial dimensionality reduction: given a spatiotemporal dataset, the spatial dimension is first reduced using the SKATER clustering algorithm, which partitions spatial units into clusters according to similarity in selected covariates.

2. Construction of cluster-level surrogate variables: the response variables and covariates are then aggregated within each cluster to produce surrogate variables representing cluster-level spatiotemporal observations, thereby replacing the original unit-level measurements with a lower-dimensional representation.

3. Learning a spatial dependence graph: a subset of surrogate covariates is selected to characterize spatial dependence, and an undirected graph is estimated over the cluster-level data obtained in step (2). Each cluster is represented by a graph vertex located at the cluster centroid. The spatial dependence structure is learned using graphical LASSO within the HUGE framework, producing a sparse graphical representation of conditional dependencies between clusters.

4. Spectral graph filtering: the estimated adjacency matrix is transformed into a graph Laplacian, after which a low-pass spectral filter is applied. The resulting filtered Laplacian is denoted by $\widetilde{L}^2$ and is constructed following the graph Laplacian filtering framework of [11].

5. Construction of the spatiotemporal covariance operator: the filtered spatial operator $\widetilde{L}^2$ is combined with a temporal Gram matrix $K$ constructed from surrogate time-series covariates. This combination is achieved through a Kronecker product, $\hat{\Sigma} = K \otimes \widetilde{L}^2$ yielding a separable spatiotemporal covariance matrix that defines the dependence structure of a Gaussian process (GP) component within the proposed log-Gaussian Cox process (LGCP) model defined over the graph vertices.

6. Model estimation: the resulting LGCP model is estimated using Integrated Nested Laplace Approximation (INLA), which enables efficient approximate Bayesian inference for latent Gaussian models.

Fig 1 summarizes the full modeling pipeline. The procedure begins with spatial clustering using SKATER to reduce dimensionality and identify regions with similar socioeconomic characteristics. Cluster-level surrogate variables are then constructed and used to estimate a spatial dependence graph via graphical LASSO. This graph is transformed into a filtered Laplacian that encodes spatial smoothness. Temporal dependencies are modeled using kernel regression based on

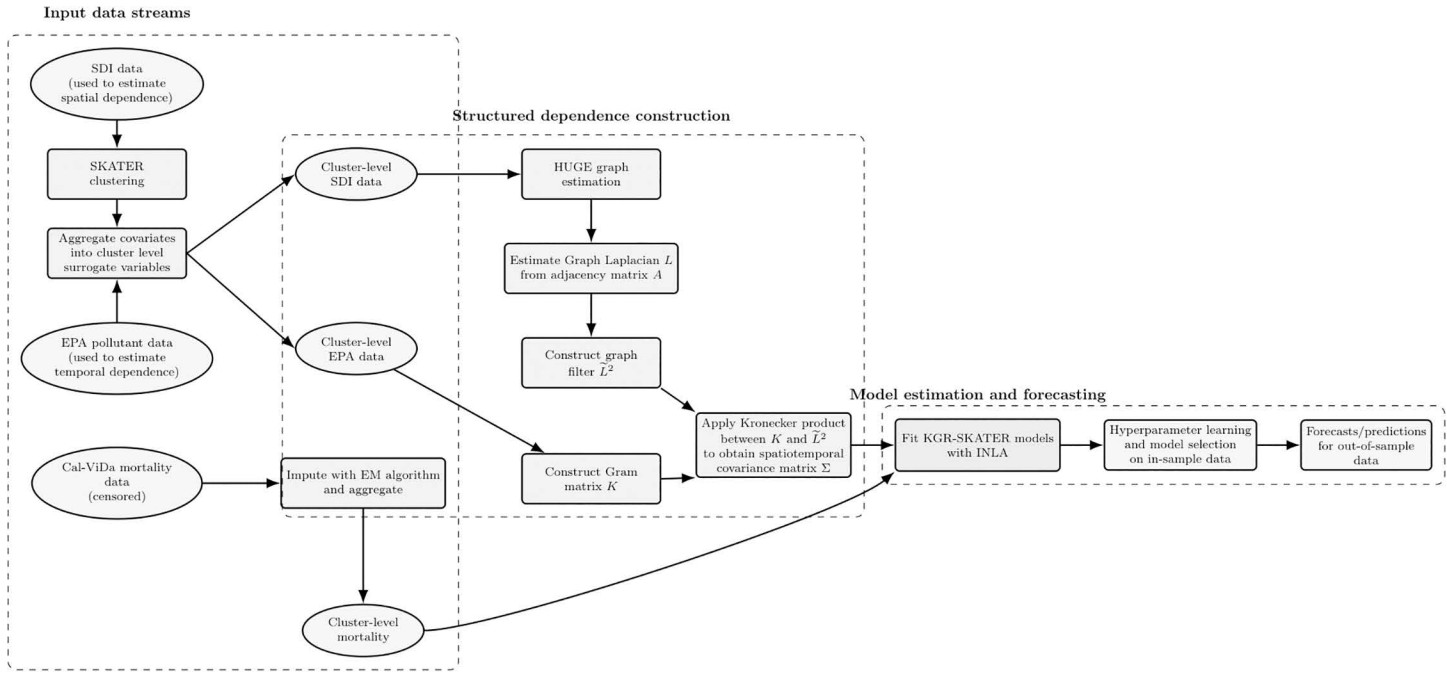

**Fig 1. Unified KGR-SKATER workflow.** County-level socioeconomic, pollution, and mortality data are first transformed into cluster-level surrogate representations. The spatial dependence structure is learned from cluster-level SDI variables using graphical model estimation and spectral graph filtering to obtain $\widetilde{L}^2$, while temporal dependence is learned from cluster-level pollution covariates through a kernel Gram matrix $K$. These structured components are combined within the KGR-SKATER model and estimated using INLA, followed by hyperparameter learning, model selection, and out-of-sample forecasting.

surrogate time series covariates. Finally, the spatial and temporal structures are combined via a Kronecker product to form a separable spatiotemporal covariance matrix used in a latent Gaussian Cox process model estimated via INLA.

## 2.1 Notation

Given an $N \times T$ data matrix $\mathbf{Y}$, with $Y_{i,t} := Y_t(\mathbf{s}_i) \in \mathbb{N}$, comprising a spatiotemporal dataset with $N$ spatial units over $T$ time steps. Here, $\mathbf{s}_i = (lat_i, long_i)$ represent latitudinal and longitudinal angles associated with the $i$-th observation. Additionally, associated time series regression covariates are observed at each location $\mathbf{s}_i$ given by $\mathbf{X}_{i,t} := \mathbf{X}_t(\mathbf{s}_i) \in \mathbb{R}^d$.

Spatial coordinates are defined using latitude and longitude values. Distances between spatial units are computed using great-circle distances derived from geographic coordinates, ensuring that spatial relationships reflect the curvature of the Earth's surface.

It will be assumed that the process being modeled will be sufficiently high-dimensional that it can be beneficial to reduce the dimensionality of the problem by mapping the spatial problem to a graph topology comprised of $C$ total units, in which the graph vertex will represent the centroid of a spatial clustering with $C = |\{\mathcal{C}_1, \mathcal{C}_2, \ldots, \mathcal{C}_C\}|$ spatial clusters. The observation matrix will then be transformed by aggregating the counts in each cluster $c \in \{1, \ldots, C\}$ to new counts denoted as $\widetilde{Y}_{c,t}$ which will then make up an aggregated count data matrix of dimension $C \times T$ denoted by $\widetilde{\mathbf{Y}}$. Equivalently, in each spatial cluster, a set of surrogate variables will be constructed for time series regression covariates in cluster $c$ denoted by $\widetilde{S}_{c,t}$. Finally, there will also be assumed to be random effects that will be incorporated in the modeling, denoted by a spatiotemporal random effect $C \times T$ matrix of latent variables denoted by $\mathbf{F}$.

The graph structure defining spatial dependence between cluster centroids is denoted by $\mathcal{G} = (\mathcal{V}, \mathcal{E})$ consisting of $C$ vertices and an edge set $\mathcal{E}$ connecting vertex pairs. There is an edge connecting vertices $v_i$ and $v_j$ if centroids for clusters $\mathcal{C}_i$ and $\mathcal{C}_j$ are associated. The edge relationships are encoding in the adjacency matrix $\mathbb{A}$, such that $A_{ij} = A_{ji}$ represents the presence of an undirected edge between nodes $i$ and $j$, i.e., $(i,j) \in \mathcal{E}$ implies $A_{ij} = 1$ and $(i,j) \notin \mathcal{E}$ implies $A_{ij} = 0$.

For the kernel regression structures used in the KGR-SKATER model approach, let $k : \mathbb{R}^d \times \mathbb{R}^d \to \mathbb{R}$ be the Reproducing Kernel Hilbert Space (RKHS) kernel characterizing the covariance operator introduced in the paper. There will be several kernel structures explored, but one choice that will be a core component that is used throughout the manuscript will be the locally periodic kernel:

$$k_{lp}(\mathbf{x}, \mathbf{x}'; \boldsymbol{\theta}) := k_p\left(\mathbf{x}, \mathbf{x}'; \sigma^2, \rho_p\right) k_{rbf}\left(\mathbf{x}, \mathbf{x}'; \sigma^2, \rho_{rbf}\right)$$

$$= \sigma^2 \exp\left(-\frac{2\sin^2(\pi |\mathbf{x} - \mathbf{x}'|/12)}{\rho_p}\right) \exp\left(-\frac{\sum_{i=1}^d |\mathbf{x} - \mathbf{x}'|^2}{2\rho_{rbf}}\right)$$

If a kernel matrix is constructed using the kernel function, it will be denoted by the $T \times T$ kernel Gram matrix $\mathbf{K}$, capturing the temporal dependence.

In the kernel definition, the variables $\mathbf{x}$ and $\mathbf{x}'$ represent vectors of surrogate covariates rather than spatial coordinates. The Euclidean distance $\|\mathbf{x} - \mathbf{x}'\|$ therefore measures similarity between covariate vectors rather than geographic proximity. Prior to kernel construction, the covariates are standardized to ensure that the distance metric is not dominated by variables with larger numerical scales.

## 2.2 Spatial clustering with SKATER

It will be assumed that in the modeling of this data, it will be sufficiently high-dimensional that it can be beneficial to reduce the dimensionality of the problem by mapping the spatial problem to a graph topology comprised of $C$ total units, in which the graph vertex will represent the centroid of a spatial clustering with $C = |\{\mathcal{C}_1, \mathcal{C}_2, \ldots, \mathcal{C}_C\}|$ spatial clusters.

 

The Spatial 'K'luster Analysis by Tree Edge Removal (SKATER) algorithm is a clustering technique designed to partition spatially structured data into homogeneous regions. It achieves this by leveraging graph theory and spatial adjacency relationships, making it particularly useful in geographical and spatial analysis. The SKATER algorithm uses a minimum spanning tree (MST) to identify clusters by progressively removing edges from the tree. The MST ensures that spatially adjacent regions are connected in a way that minimizes a specified dissimilarity measure. By removing edges with high dissimilarity, the algorithm isolates groups of nodes (spatial units) into distinct clusters.

There are three required inputs:

- Spatial Units: spatial regions (e.g., neighborhoods, districts, counties) to be clustered.

- Attribute Data: variables associated with each spatial unit (e.g., income, population density, environmental variables).

- Spatial Adjacency Matrix: neighborhood structure to identify which spatial units are neighbors, typically constructed using either direct contiguity relationships or via a distance threshold, which can be a weighted adjacency or a binary graph adjacency.

The objective is then to partition a graph $\mathcal{G}$ into $C$ disjoint spatial clusters $\mathcal{G}_1, \ldots, \mathcal{G}_C$, where their union is $\mathcal{G}$, and each is a connected subgraph. To understand how SKATER works, one starts with a minimum spanning tree as defined in [12]. The MST is built using the spatial adjacency matrix and a selected dissimilarity measure. In this tree, the nodes are still spatial units and the edges are adjacency relationships weighted by the dissimilarity between nodes. The MST is then a subgraph that has the property that it connects all nodes, it has no cycles, and it minimizes the total edge weight. There are numerous algorithms that can be used to construct the MST, see [23,24].

Having obtained the MST, the pruning exercise of SKATER involves iterative edge removal, which is performed in order to create the desired number of clusters. Each iteration of the pruning process involves identifying which edge to remove based on the edge with the highest weight (greatest dissimilarity) that still maintains the desired number of clusters. After removing an edge, the MST is then subsequently split into two disconnected subgraphs, each representing a cluster. At successive iterations, these partitions are iteratively refined on each of the previous iterations' subgraphs until a stopping criterion is satisfied. In this case, the process stops once the $k$ subgraphs, i.e., $C$ clusters, are created.

One can formalize this method of iterative pruning of the MST by solving at each iteration step the optimization program:

$$\underset{\Pi}{\arg\min} \, Q\left(\Pi\right) = \sum_{i=1}^{k} SSD_i$$

where $\Pi$ is a partition of objects into $k$ sub-trees, $Q(\Pi)$ is the cost associated with the quality of a partition, and $SSD_i$ is the sum of squares deviation in region $i$ given by

$$SSD_k = \sum_{j=1}^{m} \sum_{i=1}^{n_k} \left(\widetilde{s}_{ij} - \bar{s}_j\right)^2$$

where $n_k$ is the number of spatial objects in tree $k$, $\widetilde{s}_{ij}$ is the $j$-th attribute of spatial object $i$, $m$ is the number of attributes being used, and $\bar{s}_j$ is the average of the $j$-th attribute for all objects in tree $k$.

Therefore, formally, each iteration of SKATER proceeds by starting with a graph $G^*$ representing a minimum spanning tree $T_0$ with edges $e = 1, \ldots, E$, then at each iteration, SKATER identifies the best arrangement $S_e^T$ which is produced by removing edge $e$ from tree $T$, creating two disjoint trees $T_a$ and $T_b$

$$\arg\max_e f\left(S_e^T\right) = SSD_T - \left(SSD_{T_a} + SSD_{T_b}\right)$$

The implementation of SKATER optimization utilized in this work is based on the SKATER algorithm as proposed by Assuncao et al. in R [12,25]. This function can include one of two clustering constraints: a minimum population constraint with respect to one of the features in the spatial dataframe, or a minimum (or maximum or both) number of units being put into each cluster.

## 2.3 Producing surrogate spatial cluster covariates

Once the clustering of the spatial units is identified, the data must be aggregated to the same resolution, i.e., from the original spatial units to a less granular cluster spatial units to obtain the surrogate variables generically denoted by process $\widetilde{S}_{c,t}$ for each cluster unit $c \in \{1, \ldots, C\}$ at each time point $t \in \{1, \ldots, T\}$. This is achieved by constructing surrogate variables for each cluster unit.

The approach adopted is based on the method outlined in [26], where each covariate is aggregated from the original spatial unit to the cluster spatial unit via a population weighted mean (see Eq 1). Furthermore, the weights can be time invariant or changed according to a user specified window period, such as monthly, seasonally, annually, or upon each census.

In each spatial cluster, a set of surrogate variables will be constructed as follows for time series regression covariates in cluster $c$ given by

$$\widetilde{S}_{c,t} := \frac{\sum_{i=1}^{N} w_c \boldsymbol{X}_{i,t} \mathbb{I}\left[\boldsymbol{s}_i \in \mathcal{C}_c\right]}{\sum_{i=1}^{N} \mathbb{I}\left[\boldsymbol{s}_i \in \mathcal{C}_c\right]} \in \mathbb{N}$$

(1)

where $w_c$ represents the weight for cluster $c$. The choice of weighting mechanism for the construction of the cluster surrogate variables will be described in more detail in the following sections. These surrogate variables $\widetilde{S}_{c,t}$ will then form the graph regression covariates that will be carried into the graph and kernel estimation steps to characterize a spatiotemporal dependence structure at the cluster spatial unit.

The graph regression response variable per cluster unit $\widetilde{Y}_{c,t}$ must also be obtained, and this can be done in different ways depending on the context of the study and specified modeling assumptions. In the applications demonstrated, the observation matrix will be transformed by aggregating the counts in each cluster $c \in \{1, \ldots, C\}$ to new counts as follows for each time point

$$\widetilde{Y}_{c,t} := \sum_{i=1}^{N} Y_{i,t} \mathbb{I}\left[\boldsymbol{s}_i \in \mathcal{C}_c\right] \in \mathbb{N}$$

which will then make up an aggregated count data matrix of dimension $C \times T$ denoted by $\widetilde{Y}$.

## 2.4 Learning spatial graphical dependence from demographic variables

Given the cluster partitions and the surrogate variables, one must then construct a spatial dependence relationship that will be utilized in the subsequent KGR-SKATER modeling framework. This involves learning a graph over the $C$ spatial cluster regions to obtain a $C \times C$ adjacency matrix. This graph can be estimated on any relevant endogenous or exogenous variable, which may be more insightful than the observed neighborhood structure in some cases. Note, this also demonstrates the advantage of the application of the spatial clustering, as now the spatial dependence graph is reduced from $N$ vertices to $C$, where typically $C << N$.

In order to estimate the spatial graph, the graphical lasso package *huge* (high-dimensional undirected graph estimation) by [27] is utilized to learn the associations between the spatial cluster units. In this work, the spatial dependence structure represents spatial associations between a set of surrogate variables representing demographic factors from an index of multideprivation. However, any set of surrogate variables at the cluster spatial unit can be utilized to learn the spatial dependence in practice.

The spatial dependence learning of the graphical adjacency matrix $\mathbb{A}$ is performed as follows via a Graphical Gaussian Lasso formulation. The objective function for graphical lasso considered performs graph estimation as a sparse penalized maximum likelihood estimation problem with a penalty when estimating the precision matrix (inverse of the covariance matrix) in order to induce sparsity.

$$\widehat{\Theta} = \underset{\Theta \in \mathbb{S}_c^+(\mathbb{R})}{\arg\max} \left( \log\det\left(\Theta\right) - \mathrm{Tr}\left(\boldsymbol{S}\left(\Theta\right)\right) - \lambda\|\Theta\|_1 \right)$$

where $\Theta = \Sigma^{-1}$ is the real positive definite precision matrix, $\boldsymbol{S}$ is the empirical covariance matrix, $\mathrm{Tr}(\cdot)$ denotes the trace, $\lambda$ is the regularization parameter, and $\|\cdot\|_1$ denotes the $L_1$ norm.

The solution to this precision matrix estimation, denoted $\hat{\Theta}$, has the interpretation that if $\hat{\Theta}_{ij} = 0$, then the $i$th and $j$th variables are conditionally independent given all other variables. As the penalty term $\lambda$ increases, the bias in the precision matrix increases, and the number of zero elements in the precision matrix increases, providing a trade-off between sparsity of the precision matrix, and therefore the connectedness of the cluster spatial units in terms of the target variable upon which matrix $\boldsymbol{S}$, is estimated, and the bias.

The package *huge* will estimate multiple graphs from which an optimal one must be identified via a model selection criterion. Based on simulations, the extended Bayes criterion (EBIC) was chosen. EBIC is proposed by [28], as given by

$$EBIC_\gamma\left(\mathcal{E}\right) = -2l\left(\widehat{\Theta}\left(\mathcal{E}\right)\right) + 4|\mathcal{E}|\ln(T) + 4\gamma\ln\left(p\right), 0 \le \gamma \le 1$$

where $l\left(\cdot\right)$ is the log likelihood, $\widehat{\Theta}\left(\mathcal{E}\right)$ is the precision matrix estimate with edge set $\mathcal{E}$ (i.e., non-zero entries corresponding to graph with edge set $\mathcal{E}$, $|\mathcal{E}|$ is the number of non-zero parameters (i.e., cardinality of the edge set), $T$ is the number of observations, $p$ is the number of variables used in the graphical Lasso estimation sample covariance matrix, and $\gamma \in [0, 1]$ is a tuning parameter that adjusts the penalty on the model complexity.

## 2.5 Constructing a spatial-temporal graph regression factor prior

This section explains how to construct a latent graphical factor model that incorporates the statistical spatial dependence from the graph learned at the units of the spatial cluster with a multiple-output time series kernel based Gaussian Process factor model. The Gaussian process filters will be smoothed according to the topology of the estimated spatial graph via a form of graph filter that uses a Graph Fourier Transform (GFT) to smooth high frequency content from the factors according to the edge relations obtained spatially.

**2.5.1 Calculating a graph filter via graph fourier transform.** The estimated spatial graph structure $\mathcal{G}$ with associated adjacency matrix $\boldsymbol{A}$, where vertices represent the cluster spatial units and edges their spatial association according to a specified set of surrogate covariates, is used to construct a GFT given by spectral decomposition of the graph Laplacian with degree matrix $\boldsymbol{D}$ and adjacency matrix $\boldsymbol{A}$ given by

$$\boldsymbol{L} := \boldsymbol{D} - \boldsymbol{A} = \boldsymbol{U}^T \Lambda \boldsymbol{U}$$

for an orthonormal basis matrix $\boldsymbol{U}$ and spectral frequencies for variation over the graph denoted by a diagonal matrix of decreasing eigenvalues $\Lambda$ corresponding to eigenvectors of columns of $\boldsymbol{U}$.

Then a Laplacian spectral filtered graph is obtained by modifying these spectral frequencies using a decreasing function $\eta(\cdot)$, such that

$$\widetilde{L} = \eta\left(U^T \Lambda U\right) = U^T \eta\left(\Lambda\right) U$$

where $\eta(\lambda_i) = \widetilde{\lambda}_i \leq \lambda_i$ for each spectral frequency $i \in \{1, \ldots, C\}$. Numerous possibilities exist for $\eta(\cdot)$ such as the inverse, exponential, or ReLu filter (see discussion in [11]). In this work, a low pass filter was used that sets the filter as follows, for a given probability threshold $q \in [0, 1]$:

$$\widetilde{\lambda}_i = \eta(\lambda_i) = \begin{cases} \lambda_i, & \text{if } i/C \leq q \\ 0, & \text{otherwise.} \end{cases}$$

To understand the effect of the filtering applied to the graph Laplacian on a set of graph valued covariates or graph latent factor processes, consider the Dirichlet energy for a graph valued process $\{f_t\}$ given by

$$D\left(f_t, L\right) := f_t^T L f_t = \sum_{(i,j) \in \mathcal{E}} A_{ij}\left(f_{i,t} - f_{j,t}\right)^2 \geq 0$$

When the signal $f_t$ is smooth across the graph, the Dirichlet energy is close to zero and exactly zero for a constant signal across the graph vertices. Thus, the graph filter smooths the Laplacian such that for all $f_t$ at all times $t$ one obtains smoother signals over the graph satisfying:

$$D\left(f_t, \widetilde{L}\right) \leq D\left(f_t, L\right).$$

This spectrally filtered graph Laplacian can be used to construct a smooth prior for a latent process over the graph that can be used in the kernel graph regression within the proposed KGR-SKATER methodology. This builds on the framework proposed in [11,15] to develop the graph regression multiple output Gaussian process prior. The latent factor multiple output Graphical Gaussian process is then given by the filtered latent signal

$$\widetilde{f}_t = \widetilde{L} f_t.$$

This can then be used to construct a latent factor Gaussian process prior given generically by

$$\widetilde{f}_t \mid S_t \sim \mathcal{GP}\left(0, K \otimes \widetilde{L}^2\right)$$

(2)

Here, $\widetilde{L}^2$ acts as a spatial covariance matrix which encodes the assumption that signals observed over a graph are likely to be smooth with respect to the underlying topology and Gram matrix $K$ encodes temporal and time series regression dependence using desired surrogate times series covariates $\{S_t\}$.

## 3 Proposed kernel graph regression spatial-temporal model: KGR-SKATER

Given observed surrogate covariate process over cluster spatial units $\widetilde{S}_{c,t}$ constructed via Eq 1, latent Gaussian process factors $F_{c,t}$ specified according to prior structure in Eq 2, and observation process $\widetilde{Y}_{c,t}$ for spatial clusters $c \in \{1, 2, \ldots, C\}$, the KGR-SKATER proposed model is given $C \times T$ observation data matrix $\widetilde{Y}$ by:

$$\widetilde{Y} \mid \Lambda, \boldsymbol{F}, \widetilde{\boldsymbol{S}} \sim Poisson(\Lambda),$$

$$\Lambda_{c,t} \mid \boldsymbol{F}, \widetilde{\boldsymbol{S}} = \exp\left( \sum_{c=1}^{C} \alpha_c \mathbb{I}\left[ c \in \mathcal{C}_c \right] + \sum_{m=1}^{12} \beta_m \mathbb{I}\left[ t \bmod m \right] + F_{c,t} \right),$$

$$\boldsymbol{F} \mid \widetilde{\boldsymbol{S}} \sim \mathcal{GP}(\boldsymbol{0}, \boldsymbol{K} \otimes \widetilde{\boldsymbol{L}}^2),$$

$$Cov(\boldsymbol{F}_{c_1,t_1}, \boldsymbol{F}_{c_2,t_2} \mid \widetilde{\boldsymbol{S}}) = \left[\boldsymbol{K}^i\right]_{t_1,t_2} \left[\widetilde{\boldsymbol{L}}^2\right]_{c_1,c_2}$$

with latent intensity model parameters $\boldsymbol{\alpha} \in \mathbb{R}^C$ for $C$ spatial clusters and $\boldsymbol{\beta} \in \mathbb{R}^m$ for $m$ temporal periods, and where prior kernel hyperparameters are given generically by $\boldsymbol{\theta} = \{\boldsymbol{\theta}_1, \ldots, \boldsymbol{\theta}_J\}$ with the vector of kernel parameters given for instance in the case of an radial basis function kernel by: $\boldsymbol{\theta}_j = \left\{ \rho_{rbf,j}, \rho_{p,j}, \sigma_j^2 \right\}$. A table of kernel and hyper parameter vectors is provided in Table 1. The kernels used in this work take the form of either an additive or multiplicative product:

$$\left[\boldsymbol{K}^{(i)}\right]_{t_1,t_2} = \begin{cases} \sum_{j=1}^{J} w_j k_j \left( \left[ t_1, \widetilde{\boldsymbol{S}}_{\cdot,t_1} \right], \left[ t_2, \widetilde{\boldsymbol{S}}_{\cdot,t_2} \right]; \boldsymbol{\theta}_j \right), & \text{if } i = 1, \\ \prod_{j=1}^{J} w_j k_j \left( \left[ t_1, \widetilde{\boldsymbol{S}}_{\cdot,t_1} \right], \left[ t_2, \widetilde{\boldsymbol{S}}_{\cdot,t_2} \right]; \boldsymbol{\theta}_j \right), & \text{if } i = 2. \end{cases} \quad (3)$$

The models that are developed under this structure are known as either additive (aKGR) or multiplicative (mKGR) Kernel Graph Regression models. The interest in this proposed model structure explicitly lies in the fact that the spatial dependence, as captured by the graph Laplacian can be developed based on endogenous covariates of relevance to the application that can be used to specify directly the spatial dependence structure in the graph regression. In this manuscript, the spatial graph structure as specified by the graph filter Laplacian is learned from the surrogate variables related to the index of deprivation. Then, this spatial dependence is combined with a regression time series kernel prior regression structure via a graph product operator, which, in this work, corresponds to a Cartesian product. Monthly and cluster fixed effects are included to translate the estimated intensity function to a corresponding region and point in time. These fixed effects do not have to be included, but their inclusion leads to much better estimates.

The simulation study section illustrates how the proposed model's specification of spatiotemporal fixed effects and a more nuanced spatiotemporal covariance matrix allows it to produce better fits than the reference models presented in Section 4, especially when the periodicity of the time series being modeled is irregular. In the application study section, a variety of KGR-SKATER models will be fit on a real dataset and they are able to produce similar fits to those of the reference models, but with better coverage.

**Table 1. KGR-SKATER model formsforms.**

| Model Index | Covariates | Kernel Components | Additive or Multiplicative Mixture Kernel |
|---|---|---|---|
| $\mathcal{M}_1$ | $t$ | $k^{(t-lp)}$ | N.A. |
| $\mathcal{M}_2$ | $\widetilde{\boldsymbol{S}}$ | $k^{(x-lp)}$ | N.A. |
| $\mathcal{M}_3$ | $t, \widetilde{\boldsymbol{S}}$ | $k^{(t-lp)}, k^{(x-lp)}$ | Multiplicative |
| $\mathcal{M}_4$ | $t, \widetilde{\boldsymbol{S}}$ | $k^{(t-lp)}, k^{(x-lp)}$ | Additive |
| $\mathcal{M}_5$ | $t, \widetilde{\boldsymbol{S}}$ | $k^{(bp-lp)}, k^{(dl-lp)}, k^{(idl-lp)}$ | Additive |

Delineates proposed KGR-SKATER models based on temporal kernel choices. The first two models are single kernel models, which are then combined in a mixture to create the next two. Note: N.A. refers to the case in which a single kernel is utilized.

## 3.1 Mixture kernel components for temporal and time series kernel regression priors

There are two types of kernel structure that will be considered when designing the autocorrelation and regression structure for the factors at each vertex of the graph. These will be temporal dependence kernels that only depend on a function of time, and the other choice of kernel will be a time series kernel. In the case of the time series kernel, a multivariate time series of exogenous regression variables, which can be specified according to or independent of the graph vertices, is used to specify a conditional correlation dynamic. This is analogous to a type of Gaussian Process factor Dynamic Conditional Correlation (DCC) model expressed on a graph. A dynamic conditional correlation specification is useful when the temporal dependence of a process changes over time rather than remaining stationary. To account for this, nonstationary kernels, such as those used in [29–31], are employed to capture nonstationarity in the process being modeled by allowing the covariance function to change conditionally on observable regression time-series covariates, as is explained in the model specification below. One may think of this as a type of GP model with a conditionally deterministic covariance function (like a GARCHX model in classical time-series). Therefore, the GP model used in this work is not actually stationary. What is stationary is the hyperparameters of the covariance function, which is distinct from the GP model being stationary. One may interpret this approach as aligned with the version of Elastic methods, see [31]. Such Elastic measures account for time distortions (e.g., shifts) or different lengths between sequences. Related methods for GP design also include [32], the Global Alignment Kernel (GAK), see [33], and the KDTW approach of [34], which are based upon the Dynamic Time Warping (DTW) method of [35].

### 3.1.1 Damped period time series kernels.
The most basic form of the temporal kernel considered is the damped period kernel given by the product of a Radial Basis Function (RBF) kernel and a damped periodic kernel:

$$k^{(t-lp)}\left(t, t'; \theta\right) := k^{(p)}\left(t, t'; \sigma^2, \rho_p\right) k^{(rbf)}\left(t, t'; \sigma^2, \rho_{rbf}\right)$$

$$= \sigma^2 \exp\left(-\frac{\sin^2(\pi\,|t-t'|\,/12)}{\rho_p}\right) \exp\left(-\frac{|t-t'|^2}{2\rho_{rbf}}\right)$$

This kernel can also be specified as a time series kernel based on exogenous regression variables obtained from the surrogate time series in a generalized form, including temporal and time series surrogate covariate time series.

$$k^{(x-lp)}\left(\left[t_1, \widetilde{S}_{\cdot,t_1}\right], \left[t_2, \widetilde{S}_{\cdot,t_2}\right]; \theta\right)$$
$$:= k^{(p)}\left(\left[t_1, \widetilde{S}_{\cdot,t_1}\right], \left[t_2, \widetilde{S}_{\cdot,t_2}\right]; \sigma^2, \rho_p\right) k^{(rbf)}\left(\left[t_1, \widetilde{S}_{\cdot,t_1}\right], \left[t_2, \widetilde{S}_{\cdot,t_2}\right]; \sigma^2, \rho_{rbf}\right)$$

### 3.1.2 Locally banded and distributed lag time series kernels.
Additional kernel structures that were considered were locally banded inhomogeneous kernel structures, where at each vertex of the spatial graph $c \in \{1, \dots, C\}$, the p-banded Kernel matrix structure is considered on the d-dimensional transformed surrogate variables denoted by $\{R_{c,i,t}\}_{i=1:d,t=1:T}$ where the transformed variables are obtained as the residual of the surrogate variable after removing a structural decomposition of each vertex time series into trend $\{T_{c,i,t}\}$, seasonal $\{P_{c,i,t}\}$ and residual $\{R_{c,i,t}\}$ given by

$$R_{c,i,t} = \widetilde{S}_{c,i,t} - \widetilde{T}_{c,i,t} - \widetilde{P}_{c,i,t}$$

This decomposed residual covariate was then used in a local kernel of order $p$ given by

$$k^{(bp-lp)}\left(R_{c,i,t_1}, R_{c,i,t_2}\right) = k^{(lp)}\left(R_{c,i,t_1}, R_{c,i,t_2}; \sigma^2, \rho_p\right) \mathbb{I}\left\{|t_1 - t_2| \leq p\right\}$$

(4)

to capture residual autocorrelation structure not captured by the kernels $k^{(lp)}$. This form of local-p kernel produces a banded but dynamic kernel matrix that can be incorporated with other kernels as per Eq 3. This localized kernel structure also allows for the development of inhomogeneous banded lag structures in a localized kernel that can be designed for a desired lag pattern. By specifying important lags given by $d \in \mathcal{D} := \{0, d_1, d_2, \ldots, d_D\}$, a banded kernel at those distributed lags is given by

$$k^{(dl-lp)}\left(R_{c,i,t_1}, R_{c,i,t_2}\right) = k^{(lp)}\left(R_{c,i,t_1}, R_{c,i,t_2}; \sigma^2, \rho_p\right) \mathbb{I}\left\{|t_1 - t_2| \in \mathcal{D}\right\} \tag{5}$$

Such localized kernels as $k^{loc-p}$ will produce a kernel matrix with a lagged banded structure that can be incorporated with other kernels as per Eq 3.

**3.1.3 Instantaneous and lagged covariate interaction time series kernels.** This last class of time series kernels is designed to provide non-linear interaction effects from the surrogate time series covariates, which introduce non-linear interaction effects into the construction of the graph factor in the KGR-SKATER model family. These interaction kernels seek to produce interactions between different time series regression covariates at lags, which for the $i$-th and $j$-th surrogate covariates is modeled by kernel structure:

$$k^{(idl-lp)}\left(R_{c,i,t_1}, R_{c,j,t_2}\right) = k^{(lp)}\left(R_{c,i,t_1}, R_{c,j,t_2}; \sigma^2, \rho_p\right) \mathbb{I}\left\{|t_1 - t_2| \in \mathcal{D}\right\} \tag{6}$$

## 3.2 Families of KGR-SKATER models

In this section, the model subfamilies of KGR-SKATER structure considered by combining various kernel structures into a mixture kernel within the KGR-SKATER model framework are summarized:

$\mathcal{M}_1$ and $\mathcal{M}_2$ are the most basic KGR-SKATER models. They use a single kernel, which produces a kernel Gram matrix ($K$) of dimension $T \times T$, to represent temporal dependence, albeit in different ways. These kernels, along with the three kernels presented in Section 3.1, can be combined together with either element wise multiplication or addition to get a more nuanced temporal dependence characterization. These lead to model subfamilies $\mathcal{M}_3$, $\mathcal{M}_4$, and $\mathcal{M}_5$. Intuitively, multiplying two kernels together will produce a kernel that has high values when both of the two base kernels have high values, while adding two kernels together will produce a kernel that has high values when either one or both of the base kernels have high values. This will give different structural relations between the time series regression covariates in the kernel Gram matrix construction.

Each of these mixture kernels produces a kernel Gram matrix that is of dimension $T \times T$ because element wise operations are used. In this paper, uniform weights ($w_j = 1 \forall j$) in the multiplicative mixtures and ($w_j = \frac{1}{J}$) for the additive mixtures are considered. Once again, these different kernels lead to different temporal dependence characterizations which, when combined with the graph filter $\widetilde{L}^2$, produce different spatiotemporal dependence structure representations.

Although the locally periodic kernel provides a flexible mechanism for capturing quasi-periodic temporal dynamics, other kernel families could also be considered. For example, the Matérn kernel family introduces a smoothness parameter that allows the covariance function to interpolate between exponential and Gaussian kernels ([36]). Such kernels may provide additional flexibility in capturing varying degrees of temporal smoothness. Similarly, non-separable spatiotemporal kernels such as those proposed by [37] based on harmonic oscillator models could offer an alternative to the separable covariance structure adopted in the present study.

## 4 Benchmark reference models

This section presents benchmark reference models that will be compared with those produced by the proposed modeling framework. These reference models were chosen because they all have a GP component like

KGR-SKATER models and build upon each other, culminating in an LGCP. A key component is a latent factor GP model that serves as the foundation for many popular models, including latent Gaussian models, which are used to model non-Gaussian responses. In spatial (or spatiotemporal) data settings, they are commonly used as random effects, like in a spatial GLMM, or to model the intensity function of a stochastic process, like in a Log Gaussian Cox Process (LGCP).

The first reference model is a spatial GLMM, which is used to model spatially dependent non-Gaussian variables in geostatistical contexts [5]. This reference model assumes the observed data conditionally follows a Poisson distribution and the intensity process $\Lambda_{c,t}$ at each cluster region can be modeled using a mixed effects model with a log link.

**Reference model 1: Poisson GLMM ($\mathcal{M}_1^R$)**

$$\widetilde{\boldsymbol{Y}} \mid \Lambda, \beta, \boldsymbol{F} \sim Poisson(\Lambda),$$

$$\Lambda_{c,t} = \exp\left(\sum_{c=1}^{C} \alpha_c \mathbb{I}\left[c \in \mathcal{C}_c\right] + \sum_{m=1}^{12} \beta_m \mathbb{I}\left[t \bmod m\right] + F_c\right)$$

$$\boldsymbol{F} \mid \tau \overset{i.i.d.}{\sim} MVN(\boldsymbol{0}, \tau\Sigma)$$

The $\mathcal{M}_1^R$ reference model contains an intercept for each cluster, monthly fixed effects, and a random effect for each location which is estimated as a spatial random process that is assumed to be an unobservable stationary Gaussian random vector with spatial correlation captured by a homoskedastic covariance matrix $\Sigma$. As a baseline, a diagonal covariance matrix for $\Sigma$ is used to allow for a reference that treats the regression as a type of Seemingly Unrelated Regression (SURE) LGCP framework. This is the most basic reference model to be considered. The hyperparameter $log(\tau)$ is assigned a vague log gamma prior $log\,\Gamma(1, 0.00001)$.

Obviously, this model is naive because spatial random effects are unlikely to be iid. The second reference model ($\mathcal{M}_2^R$), a Besag-York-Mollie (BYM) model, adds some spatial dependence structure. A BYM model is a log-normal Poisson model with an intrinsic conditional autoregressive (ICAR) component to capture spatial autocorrelations, i.e., a Besag model, plus a standard random effects component to capture non-spatial heterogeneity.

**Reference model 2: Besag-York-Mollie Model ($\mathcal{M}_2^R$)**

$$\widetilde{\boldsymbol{Y}} \mid \Lambda, \beta, \boldsymbol{F}, \widetilde{\boldsymbol{S}} \sim Poisson(\Lambda),$$

$$\Lambda_{c,t} \mid \widetilde{\boldsymbol{S}} = \exp\left(\sum_{c=1}^{C} \alpha_c \mathbb{I}\left[c \in \mathcal{C}_c\right] + \sum_{m=1}^{12} \beta_m \mathbb{I}\left[t \bmod m\right] + \phi_c + F_c\right),$$

$$\boldsymbol{F} \mid \widetilde{\boldsymbol{S}}, \tau \sim MVN(\boldsymbol{0}, \tau\Sigma,$$

$$p(\phi) \propto \exp\left(-\frac{1}{2}\sum_{c_1 \sim c_2}(\phi_{c_1} - \phi_{c_2})^2\right)$$

Reference model ($\mathcal{M}_2^R$) can be seen as a more general form of reference model ($\mathcal{M}_1^R$), where instead of assuming that random effects between spatial units are iid, the BYM model imposes an ICAR spatial dependence structure. Recall that ICAR components are conditionally normally distributed. It also has an additional standard normal random effect to capture any residual variation that is not spatially dependent.

The third reference model ($\mathcal{M}_3^R$) takes the form of an LGCP model. LGCPs are applicable to a variety of spatio-temporal settings mainly because they are flexible and relatively tractable. They provide good predictions but are not very interpretable [38]. For reference model 3, a locally periodic temporal kernel is used as the covariance of the underlying GP.

## Reference model 3: LGCP with time kernel ($\mathcal{M}_3^R$)

$$\widetilde{Y} \mid \Lambda, F, \beta \sim Poisson(\Lambda),$$

$$\Lambda_{c,t} = \exp\left(\sum_{c=1}^{C} \alpha_c \mathbb{I}\left[c \in \mathcal{C}_c\right] + F_t\right),$$

$$F \mid \theta \sim \mathcal{GP}(0, K) \text{ where } \theta = \{\rho_{rbf,r3}, \rho_{p,r3}, \sigma^2\},$$

$$K = \left[k_{lp}(t_1, t_2; \sigma^2, \rho_p, \rho_{rbf})\right]_{t_1=1:T, t_2=1:T}$$

## 5 Estimation via integrated nested laplace approximation (INLA)

In order to estimate the counts $\widetilde{Y}_{c,t}$ for all time points $t \in \{1, \ldots, T\}$ at all units $c \in \{1, \ldots, C\}$, the intensity for each cluster and time point $\Lambda_{c,t}$ must be estimated. This involves the estimation of the model intensity function regression parameters $\beta = \{\beta_1, \ldots, \beta_{11}, \beta_{c1}, \ldots, \beta_C\}$, the spatial temporal factors $F = \{F_{1,1}, \ldots, F_{1,T}, F_{2,1}, \ldots, F_{C,T}\}$, and hyperparameters of the Gaussian process $\theta$ that will depend on the kernel functions used. INLA significantly speeds up the estimation and inference of latent Gaussian models and thus can be leveraged to rapidly compare many different model structures and choices against each other for tasks like kernel hyperparameter learning.

In general, INLA can perform hyperparameter learning; however, in the KGR-SKATER model, the use of the graph filter combined with the Gaussian process kernel matrix means that the standard INLA package will not accommodate this formulation directly. This can be resolved in the INLA approach while still using the standard INLA packages in R via a grid search for the estimation of $\theta$ with optimal values selected based on model ranking criteria such as DIC or WAIC. Essentially, for each point on the hyperparameter grid, the proposed LGCP model is fit in INLA on an in sample dataset and ranked by the resulting information criterion. Each model can be fit with INLA quickly, but a grid search with many hyperparameters can become computationally expensive. To alleviate some of the computational burden, a grid was constructed using Latin hypercube sampling, allowing for fewer grid points whilst adequately covering the parameter space of the hyperparameters.

For each set of hyperparameters, the INLA method is adopted to estimate the marginal posteriors of each parameter of interest in Eq 9. This is done in R using the library *INLA* and *brinla* libraries. See S7 Appendix for a demonstration of how $\mathcal{M}_1^R$ and $\mathcal{M}_2^R$ and an LGCP model are fit on synthetic data. INLA is a useful tool for conducting approximate Bayesian inference in cases, like Log Gaussian Cox Processes, where traditional Bayesian approaches like MCMC are more difficult to implement. Even though MCMC methods produce asymptotically more accurate results, they are much slower in comparison to INLA.

The main steps of INLA proposed in [8] are to use the joint density of the latent field, hyperparameters, and data given for reference models $\mathcal{M}_1^R$ and $\mathcal{M}_2^R$ by:

$$\pi(\Lambda, \beta, diag(\sigma_1, \ldots, \sigma_c) \mid \widetilde{Y}) \propto \pi(\beta)\pi(\sigma_1) \ldots \pi(\sigma_c)\pi(\Lambda \mid \beta, diag(\sigma_1, \ldots, \sigma_c))$$

$$\prod_{t=1}^{T} \pi(\widetilde{y}_t \mid \Lambda_t, \beta, diag(\sigma_1, \ldots, \sigma_c)), \tag{7}$$

and for KGR-SKATER models by:

$$\pi(\Lambda, \beta \mid \widetilde{Y}, \widetilde{S}, (\hat{K} \otimes \widetilde{\hat{L}}^2)) \propto \pi(\beta)\pi(\Lambda \mid \beta, (\hat{K} \otimes \widetilde{\hat{L}}^2)) \prod_{t=1}^{T} \pi(\widetilde{y}_t \mid \Lambda_t, \beta, \widetilde{S}, (\hat{K} \otimes \widetilde{\hat{L}}^2)) \tag{8}$$

where Eq 7 corresponds to the estimation of $\mathcal{M}_1^R$ and $\mathcal{M}_2^R$ and Eq 8 corresponds to the estimation of the KGR-SKATER models. Note that for $\mathcal{M}_1^R$ and $\mathcal{M}_2^R$, $\Sigma$ is a diagonal matrix $diag(\sigma_1, \ldots, \sigma_c)$ and for the proposed KGR-SKATER models $\Sigma = (\hat{\boldsymbol{K}} \otimes \widetilde{\hat{\boldsymbol{L}}}^2)$.

Notice that since the latent factor covariance $(\hat{\boldsymbol{K}} \otimes \widetilde{\hat{\boldsymbol{L}}}^2)$ of KGR-SKATER models is estimated prior to the INLA step, it is included in the conditioning statement in the steps below for how INLA estimates KGR-SKATER models. Also note that $\widetilde{\boldsymbol{S}}$ denotes the collection of surrogate spatial cluster covariates, and these are used to estimate $\hat{\boldsymbol{K}}$ and $\widetilde{\hat{\boldsymbol{L}}}^2$). Since all of the distributions from this point on in this section will already be conditioned on $\hat{\boldsymbol{K}} \otimes \widetilde{\hat{\boldsymbol{L}}}^2$), $\widetilde{\boldsymbol{S}}$ will be omitted for simplicity of notation.

The INLA approximation for KGR-SKATER models proceeds under two generic steps as follows: 1. Approximate the posterior of the hyperparameters $\pi(\beta \,|\, \widetilde{\boldsymbol{Y}}, (\hat{\boldsymbol{K}} \otimes \widetilde{\hat{\boldsymbol{L}}}^2))$ as

$$\widehat{\pi}(\beta \,|\, \widetilde{\boldsymbol{Y}}, (\hat{\boldsymbol{K}} \otimes \widetilde{\hat{\boldsymbol{L}}}^2)) \propto \frac{\pi(\Lambda, \beta \,|\, \widetilde{\boldsymbol{Y}}, (\hat{\boldsymbol{K}} \otimes \widetilde{\hat{\boldsymbol{L}}}^2))}{\pi_G(\Lambda \,|\, \beta, (\hat{\boldsymbol{K}} \otimes \widetilde{\hat{\boldsymbol{L}}}^2), \widetilde{\boldsymbol{Y}})} \Big|_{\Lambda = \mu(\beta)}$$

where $\pi_G(\Lambda \,|\, \beta, (\hat{\boldsymbol{K}} \otimes \widetilde{\hat{\boldsymbol{L}}}^2), \widetilde{\boldsymbol{Y}})$ is a Gaussian approximation of the full conditional distribution of $\Lambda$ and $\mu(\beta)$ represents the mode of the conditional distribution for given values of the hyperparameters and a Cholesky decomposition is used for the precision matrix, i.e.,

$$\pi_G(\Lambda \,|\, \beta, (\hat{\boldsymbol{K}} \otimes \widetilde{\hat{\boldsymbol{L}}}^2), \widetilde{\boldsymbol{Y}}) \sim N(\mu(\beta), Q_\Lambda^{-1}(\beta)) \text{ and}$$

$$\pi_G(\Lambda_j \,|\, \beta, (\hat{\boldsymbol{K}} \otimes \widetilde{\hat{\boldsymbol{L}}}^2), \widetilde{\boldsymbol{Y}}) \sim N(\mu_j(\beta), (Q_\Lambda^{-1}(\beta))_{jj})$$

2. Then approximate the marginal posteriors of each parameter of interest $\Lambda$ using numerical integration as follows:

$$\widehat{\pi}(\Lambda | \widetilde{\boldsymbol{Y}}) = \int \pi_G(\Lambda \,|\, \beta, (\hat{\boldsymbol{K}} \otimes \widetilde{\hat{\boldsymbol{L}}}^2), \widetilde{\boldsymbol{Y}}) \pi_G(\beta \,|\, (\hat{\boldsymbol{K}} \otimes \widetilde{\hat{\boldsymbol{L}}}^2), \widetilde{\boldsymbol{Y}}) d\beta$$

$$\approx \sum_{k=1}^{K} \pi_G(\Lambda \,|\, \beta_k, (\hat{\boldsymbol{K}} \otimes \widetilde{\hat{\boldsymbol{L}}}^2), \widetilde{\boldsymbol{Y}}) \pi_G(\beta_k \,|\, (\hat{\boldsymbol{K}} \otimes \widetilde{\hat{\boldsymbol{L}}}^2), \widetilde{\boldsymbol{Y}}) \delta_k$$

$$(9)$$

where there are K integration points $\beta_k$ and area weights $\delta_k$ defined by some numerical integration scheme

In [8], extensions of this approximation approach are developed based on the Laplace and simplified Laplace approximations as alternative methods to obtain better approximations for $\pi(\Lambda_j \,|\, \beta, (\hat{\boldsymbol{K}} \otimes \widetilde{\hat{\boldsymbol{L}}}^2), \widetilde{\boldsymbol{Y}})$. These are especially useful in non-Gaussian likelihood cases like Poisson processes. The Laplace approximation is given by:

$$\pi_{LA}(\Lambda_j \,|\, \beta, (\hat{\boldsymbol{K}} \otimes \widetilde{\hat{\boldsymbol{L}}}^2), \widetilde{\boldsymbol{Y}}) \propto \frac{\pi(\Lambda, \beta \,|\, \widetilde{\boldsymbol{Y}})}{\widetilde{\pi}_{GG}(\Lambda_{-j} \,|\, \Lambda_j, \beta, (\hat{\boldsymbol{K}} \otimes \widetilde{\hat{\boldsymbol{L}}}^2), \widetilde{\boldsymbol{Y}})} \Bigg|_{\Lambda_{-j} = \Lambda_{-j}^*(\Lambda_j, \beta)}$$

where $\widetilde{\pi}_{GG}(\Lambda_{-j} \,|\, \Lambda_j, \beta, (\hat{\boldsymbol{K}} \otimes \widetilde{\hat{\boldsymbol{L}}}^2), \widetilde{\boldsymbol{Y}})$ represents an the Gaussian approximation of $\pi(\Lambda_{-j} \,|\, \Lambda_j, \beta, (\hat{\boldsymbol{K}} \otimes \widetilde{\hat{\boldsymbol{L}}}^2), \widetilde{\boldsymbol{Y}})$, which is a different conditional density than $\pi_G(\Lambda \,|\, \beta, (\hat{\boldsymbol{K}} \otimes \widetilde{\hat{\boldsymbol{L}}}^2), \widetilde{\boldsymbol{Y}})$, and $\Lambda_{-j}^*(\Lambda_j, \beta)$ is the mode. This approximation has optimal performance but requires the largest computational budget since it must be computed for each value of $\Lambda_{-j}$, see discussion in [39]. Therefore, there is also a simplified Laplace approximation option, denoted by $\pi_{SLA}(\Lambda_j \,|\, \beta, (\hat{\boldsymbol{K}} \otimes \widetilde{\hat{\boldsymbol{L}}}^2), \widetilde{\boldsymbol{Y}})$, which is obtained via a Taylor series expansion of $\pi_{LA}(\Lambda_j \,|\, \beta, (\hat{\boldsymbol{K}} \otimes \widetilde{\hat{\boldsymbol{L}}}^2), \widetilde{\boldsymbol{Y}})$ around $\Lambda_j = \mu_j(\beta)$. This approximation is sufficiently accurate in many applications and requires a smaller computational budget, see [39].

Not only does INLA quickly provide approximations for the marginal posterior distributions of all the parameters of interest, but it can also generate approximations of the posterior predictive distribution, useful for forecasting:

$$\pi\left(\boldsymbol{Y}_{t+1},\ldots,\boldsymbol{Y}_{t+h}\,|\,\boldsymbol{y}_{1:t}\right) = \int \pi\left(\boldsymbol{Y}_{t+1,\ldots,t+h}\,|\,\boldsymbol{\Lambda}\right)\pi\left(\boldsymbol{\Lambda}\,|\,\boldsymbol{y}_{1:t}\right)d\boldsymbol{\Lambda}$$

$$\approx \sum_{k=1}^{K} \pi\left(\boldsymbol{Y}_{t+1,\ldots,t+h}\,|\,\boldsymbol{\Lambda_k}\right)\widehat{\pi}\left(\boldsymbol{\Lambda}_k\,|\,\boldsymbol{y}_{1:t}\right)$$

In order to get the approximation of the predictive posterior, a quantization is used and the set of $K$ support points is chosen such that they cover all areas with non-negligible posterior density, and for each support point an estimate of the posterior density is given, see further discussion in [40].

## 6 KGR-SKATER model validation case study

Before undertaking the detailed real data case study, this section briefly outlines a synthetic data case study that was undertaken to illustrate the behavior and accuracy of the KGR-SKATER modeling framework, compared to the reference models. The bulk of the synthetic data generation, validation, and results are outlined in detail in S13 Appendix. A synthetic dataset was generated intentionally with frequency and amplitude modulated intensity functions to assess performance in a challenging setting. A BYM model akin to $\mathcal{M}_2^R$ and a KGR-SKATER model akin to $\mathcal{M}_1$ are fit to this dataset.

In a one month ahead rolling forecast exercise, the reference model simply makes the same periodic prediction repeatedly. However, the KGR-SKATER can adapt more flexibly to capture the patterns of the resulting intensity process both in terms of prediction accuracy and coverage. Details of this relative performance comparison are provided in S13 Appendix.

### 6.1 Practical implementation considerations

Applying the KGR-SKATER framework in practice involves several modeling choices. First, the selection of covariates used for spatial graph estimation should be guided by domain knowledge regarding the drivers of spatial dependence. Second, the number of spatial clusters can be selected using clustering diagnostics such as silhouette width or domain-specific constraints. Third, kernel structures should be chosen based on the temporal characteristics of the data; locally periodic kernels are appropriate when quasi-periodic dynamics are expected.

Missing data or censored observations can be handled using standard imputation procedures prior to model estimation. In this study, an expectation–maximization procedure was used to impute censored mortality counts. Finally, hyperparameters governing the kernel functions can be selected using information criteria such as DIC or WAIC.

## 7 Application study: Modeling respiratory-related mortality in California

To demonstrate the KGR-SKATER methodology, this paper will attempt to model respiratory-related deaths across California whilst incorporating socioeconomic data for spatial dependence and air quality data as spatiotemporal time series regression covariates.

### 7.1 Data

Several publicly available data sources were used in this study. In this section, a brief outline of the details of each data set is provided; for a more comprehensive discussion of the data details and some plots in addition to the ones provided below, see S1 Appendix. The socioeconomic data in the form of Social Deprivation Indices (SDIs) comes from [41] (see (https://www.soa.org/resources/research-reports/2020/us-mort-rate-socioeconomic/#excel)) and the Society of Actuaries (SoA), air quality data from the [42] (see https://aqs.epa.gov/aqsweb/documents/data_api.html), and mortality data from the [43] (see https://cal-vida.cdph.ca.gov/VSQWeb). Table 2 presents an empirical summary of the application study data:

**Table 2. Summary statistics for covariates in application study.**

| Variable | Minimum | Q1 | Median | Mean | Q3 | Maximum |
|---|---|---|---|---|---|---|
| Lead ($\mu g/m^3$) | 0.012 | 0.014 | 0.016 | 0.018 | 0.019 | 0.066 |
| CO (ppm) | 0.229 | 0.300 | 0.346 | 0.372 | 0.433 | 0.650 |
| $SO_2$ (ppb) | 0.264 | 0.486 | 0.655 | 0.645 | 0.777 | 1.132 |
| $NO_2$ (ppb) | 3.402 | 4.939 | 7.139 | 8.351 | 11.071 | 21.668 |
| $O_3$ (ppm) | 0.008 | 0.020 | 0.026 | 0.025 | 0.029 | 0.037 |
| $PM_{10}$ ($\mu g/m^3$) | 7.000 | 13.000 | 18.000 | 20.090 | 25.000 | 70.500 |
| $PM_{2.5}$ ($\mu g/m^3$) | 4.200 | 6.700 | 7.800 | 8.550 | 9.600 | 23.350 |
| AQI | 25.000 | 33.000 | 36.000 | 38.340 | 40.500 | 75.000 |
| SDI Score | 67.380 | 93.450 | 106.950 | 112.530 | 131.690 | 183.670 |

Empirical numerical summary of covariates (across all counties) for the application study. Note that ppm and ppb stand for parts per million and parts per billion respectively. The summary for mortality was not insightful enough to include (mostly 0s).

The SoA's socioeconomic dataset contains 11 different sub-indices for each county for the years 2011–2019:

The subindices in Table 3 were grouped into principal components, and the component that explained the most variation in mortality was used to create a single multidimensional index of social deprivation, SDI. This variable SDI is used in the clustering and graph estimation stages for the KGR-SKATER filtered graph Laplacian spatial dependence structure.

The air quality data provided by the EPA includes daily atmospheric measurements for the seven main air pollutants: lead, carbon monoxide (CO), sulfur dioxide ($SO_2$), nitrogen dioxide ($NO_2$), ozone ($O_3$), $PM_{10}$ and $PM_{2.5}$, and the AQI index value associated with a given pollutant for that day. Once the daily averages for each pollutant, for each county, had been queried and consolidated (see S2 Appendix for details of the procedure), it was aggregated into surrogate monthly time series regression covariates. These will be used to construct the time series regression kernel matrix for the KGR-SKATER model, denoted $\boldsymbol{K}_{EPA}$.

The observation process is comprised of mortality data categorized by cause of death from the California Department of Health (CDPH). In this case study, it was assumed that since air pollution primarily affect people's health through their respiratory systems, the raw dataset was filtered to only include "chronic lower respiratory diseases" and "influenza and pneumonia" as causes of death because these were the only two that were distinctly respiratory-related. The months of

**Table 3. Socioeconomic variables used to construct SDI score.**

| SDI subindices |
|---|
| 1. Percentage of the population aged 25 and over with less than 9 years of education |
| 2. Percentage of the population aged 25 and over with at least 4 years of college education |
| 3. Percentage of the population aged 16 and over employed in a white collar occupation |
| 4. Unemployment rate for the population 16 years and over |
| 5. Median household income adjusted for local housing costs |
| 6. Ratio of the average household income in the lowest quintile to the average household income in the highest quintile |
| 7. Percentage of the population below the federal poverty threshold |
| 8. Median home value for owner occupied units |
| 9. Median gross rent for rental units |
| 10. Percentage of housing without a telephone |
| 11. Percentage of housing without complete plumbing |

2014–2019 were selected as the study's time window in order to avoid the COVID pandemic's influence on respiratory deaths, which could confound the results. This filtering of the CDPH dataset resulted in observations of the number of deaths each month by county and by age group, e.g., 1–4 years, 15–24 years, up to 85 and older. However, this raw data selection had a known censoring that was applied by the data provider to protect privacy. Censoring was applied to any nonzero death counts less than 11 in each county and age group, in order to mask small cell counts. These censored death counts were imputed based on the Expectation Maximization (EM) Algorithm (see S3 Appendix for details).

While this approach provides statistically consistent estimates under standard missing-data assumptions, it may introduce additional uncertainty into the analysis. In particular, imputation may attenuate extreme observations and potentially underestimate variability. However, the relatively small proportion of censored observations suggests that any resulting bias is likely to be modest.

With the mortality data now available by age group, month, and county, the data across age groups was added together to create the main response variable. Next, the county level data was added together to match the desired deaths per month by cluster format needed for the model. For the purpose of this application study, the spatiotemporal effects of air quality and socioeconomic status were the main focus; hence, the age groups were aggregated into a total county-level observation in order to reduce the zero inflation and overdispersion.

Time series of the cluster level surrogate data for the SDI, AQI and aggregated mortality are shown in Fig 2 for the case of seven spatial SKATER clusters:

## 7.2  SKATER clustering

The SKATER clustering was performed using the county level SDI covariate information. Details of the procedure for deciding this are provided in the S5 Appendix. There are 58 counties in California and these will be reduced to a total number of clustered regions determined based on the cluster performance, between 2 and 10 spatial clusters obtained from the SKATER method. In the SKATER clustering package in R, one can also consider using one of the two constraints mentioned in 2.2. These constraints are useful for application studies like this, where one might want more balanced subgroups. The cluster order, i.e., the number of clustered spatial groups of counties of California, grouped according to the similarity and spatial contiguity of the Social Deprivation over time, was determined via a silhouette plot analysis. This was done in R using *fviznbclust()*, and analysis was performed to compare average widths from two to ten clusters.

The silhouette plot, which can be found in S4 Appendix, shows that the average width for two clusters is closest to 1 and that the average drops off progressively. However, it should be noted that the standard silhouette method implemented in these functions does not account for spatial contiguity adopted by SKATER clustering. In S8 Appendix, SKATER is run under the different constraints for two and seven clusters. One can see that the two cluster case is not very informative, and that running SKATER unconstrained may lead to clusters with just one or two counties. Ultimately, based on the silhouette cluster analysis, it was decided that a suitable tradeoff between spatial modeling and cluster dimension reduction was to use seven clusters. Furthermore, the minimum population constraint was used to maintain balanced clusters. These clusters are displayed in Fig 3.

After the clustering step, surrogate variables are constructed as outlined in Section 2.3. However, first, the temporal resolutions of both the raw county level SDI and EPA data have to be matched to the mortality data's temporal resolution. The data from EPA measuring stations are daily averages, so the median of each month's measurements was used to get a monthly value for each pollutant. This aggregation was done after filtering out stations with bad data. Only one SDI value is calculated for each year, so the same value was used for each month within a given year. The population weights used to calculate the weighted average for each covariate were obtained by calculating the proportion of the population within each county in a given cluster using census data for each year. Once again, the surrogate variable for the SDI data is used to construct a spatial dependence structure and the surrogate variables for the EPA data are used to construct a temporal dependence structure in the time series kernel structure.

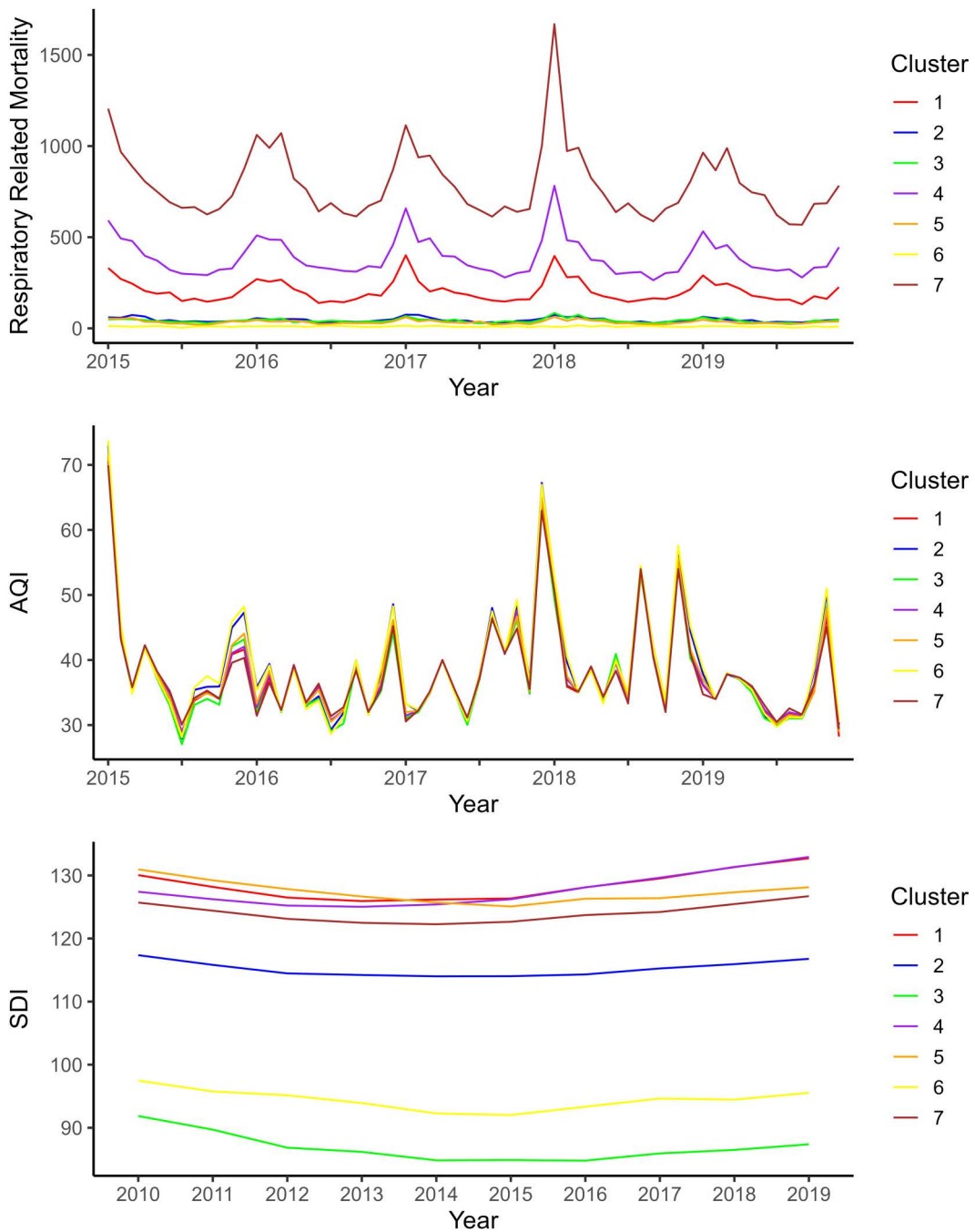

**Fig 2. Time series plots of aggregated response and aggregated surrogate time series explanatory variables for each cluster.**

### 7.3 Characterization of spatial dependence structure: Graphical LASSO

Once the spatial clustering is performed, the next step in the KGR-SKATER proposed modeling approach is to estimate the graphical spatial dependence. This is achieved with a graphical LASSO method, where the R package *huge* is utilized. The inputs are the surrogate variables from the SDIs and the resulting estimated graph complexity,

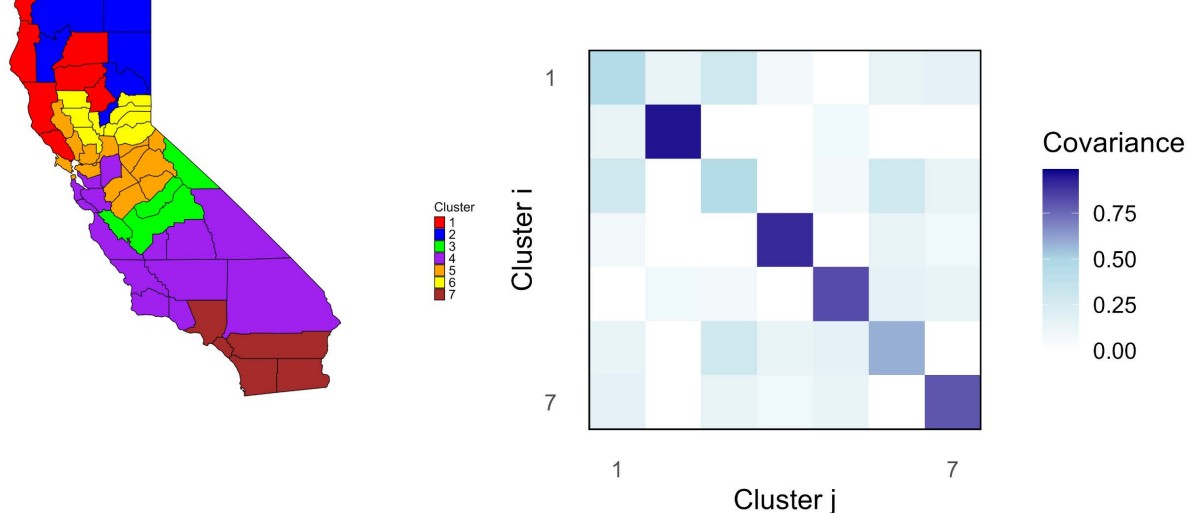

**Fig 3. Spatial dependence structure for seven clusters and a minimum population constraint visualized.** County boundary shapefiles obtained from the US Census Bureau (https://catalog.data.gov/dataset/tiger-line-shapefile-2016-state-california-current-place-state-based). These are in the public domain. Maps were generated by the authors using R packages (*maps, sf, ggplot2*).

given by the cardinality of the graph edge set determined using the EBIC model selection criterion (see further details in S6 Appendix).

After feeding the surrogate SDI variables into *huge* and following the steps outlined in Section 3.2, a graph, from which the graph filter $\widetilde{L}^2$ can be calculated, was estimated. Fig 3 below illustrates the estimated spatial dependence structure for seven clusters under the minimum population constraint. To see all cluster and graph filter results, refer to S8 Appendix.

### 7.4 Characterization of temporal dependence structure

The other component within the covariance matrix of the Gaussian process to be estimated for a KGR-SKATER model is $K$, the kernel Gram matrix that characterizes the temporal dependence structure. In Table 1, a variety of kernel choices are posited as potentially useful depending on the type of dependence structure that is hypothesized to exist. These different kernel components and mixture choices lead to the five different proposed models used in the application study to demonstrate the KGR-SKATER modeling capabilities. See S9 Appendix for the five KGR-SKATER models written out explicitly. Admittedly, these five models are similar; the main point of including all five models is to illustrate that there are many different ways to specify the covariance structure of the underlying intensity signal. One can see and compare the visual representations of the precision matrices for each proposed model in S11 Appendix. These models have the minor drawback of increased dimensionality in comparison to the simpler reference models. But in the next section, the improvement with respect to in-sample and out-of-sample fit will be exemplified.

### 8 Results

After presenting the evaluation metrics that will be used to compare in and out of sample performance, the reference and proposed models' performance in and out of sample will be discussed. This section will show that the application study's data can be modeled with several of the proposed models just as well as with the reference models. This is complemented by an increase in Frequentist coverage rates out of sample.

## 8.1 Evaluation metrics

When the reference and proposed models are fit in sample, the deviance information criterion (DIC) and Watanabe-Akaike information criterion (WAIC) are extracted and compared. WAIC results will be shown throughout the rest of the paper. For DIC results and a discussion on how these two metrics compare, see S10 Appendix.

To get an idea of how well each model fits the in sample data, the scaled root mean squared error (RMSPE) is evaluated for each cluster. Standard RMSPEs values for each cluster would be drastically different because the population sizes between clusters still vary significantly. As a result, the error rate is scaled by dividing the standard RMSPE value by the average of the actual observed data points, i.e.,

$$\text{scaled } RMSPE_c = \sqrt{\frac{\sum_{t=1}^{n}(\widetilde{Y}_{c,t} - \hat{Y}_{c,t})^2}{n_{sample}}} \Big/ \widetilde{\mu}_{c,n}$$

Since the response takes the form of counts, INLA can only predict the average intensity of this Poisson response, $\hat{\lambda}_{c,t+h}$, at each location and time point. Because of this, an estimate of the true intensity is defined as $\hat{\lambda}_{c,t+h}^{obs}$. This estimate is obtained by taking the average of the number of deaths observed for month $t+h$ over all years (2015–2019). So when the average intensity for December 2019 is to be predicted for instance, that prediction is compared with the average of observed mortalities during December 2015, 2016, 2017, and 2018.

## 8.2 In sample fitting

To estimate each of the KGR-SKATER and reference models, INLA is fit to the aggregated respiratory-related mortalities for each cluster group from 2015 to 2019. The last six months of 2019 (months 55–60) are held out for the out of sample forecasting exercise. All three reference models and five proposed models are fit on the data clustered into two and seven clusters under the minimum population constraint. Using INLA, each of the models was fit in at most a couple of minutes. Additional output, such as the posterior densities of each parameter and hyperparameter, the model's WAIC values, the posterior predictive fitted values, and uncertainty quantification, can also be extracted with ease.

From the WAIC values in Table 4, notice that as the number of clusters/spatial units increases, the WAIC increases, meaning these models of higher complexity are not as favorable. This is to be expected because the size of the graph filter $\widetilde{L}^2$ depends on the number of clusters, and this directly affects how large the covariance matrix of each model's underlying Gaussian process will be.

As shown in Figs 4 and 5, the model fits for the reference and proposed models with the smallest WAIC, $\mathcal{M}_2^R$ and $\mathcal{M}_4$, the proposed model appears to be better with respect to uncertainty quantification. The accuracy of the fits is too close to differentiate visually, so the in sample scaled RMSPEs were calculated for each model, for two and seven clusters, and

**Table 4. WAIC values for each model.**

| Reference Models | | Proposed Models | | | | |
|---|---|---|---|---|---|---|
| #1 | #2 | #1 | #2 | #3 | #4 | #5 |
| **WAIC for 2 clusters** | | | | | | |
| 1881.963 | 1881.963 | **977.525** | 980.437 | 981.191 | 978.203 | 980.494 |
| **WAIC for 7 clusters** | | | | | | |
| 4065.576 | 4065.576 | 2818.437 | 2821.594 | 2820.723 | **2808.770** | 2815.812 |

Proposed models are preferred by WAIC for both two and seven clusters. The reference models are preferred when using DIC as the model selection criterion. These results were obtained using SKATER's minimum population constraint.

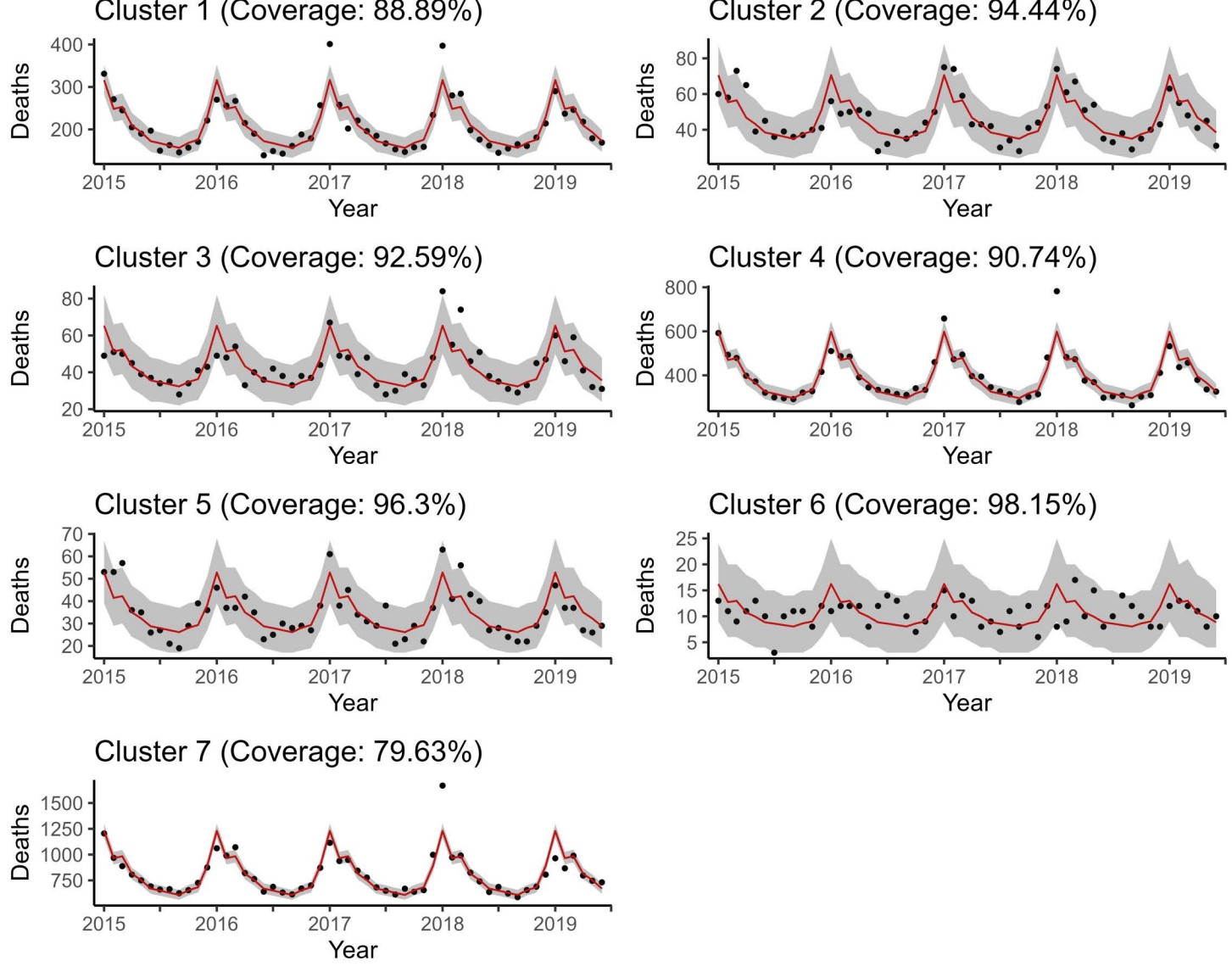

**Fig 4. Posterior predictive mean and credible interval bands estimated by $\mathcal{M}_2^R$.** Notice that the proposed models perform better with respect to uncertainty quantification and about the same with respect to in-sample fit.

put into a table. This table can be found in S12 Appendix. The reference models turn out to have slightly less bias, but not by much.

As mentioned in Section 4, the proposed models do not need monthly fixed effects like the reference models to generate acceptable fits. These fits, which can be seen in S16 Appendix, are smoother and generally less accurate because their predictions tend towards the cluster mean instead of following the seasonal patterns. Including monthly fixed effects results in model fits indistinguishable from those of $\mathcal{M}_1^R$ and $\mathcal{M}_2^R$. It turns out that due to the strong, persistent periodicity in the application study data, the monthly fixed effects capture the seasonal patterns very well. Since $\mathcal{M}_4$ had the lowest WAIC values among the proposed models, it will be carried into the out of sample fitting analysis along with $\mathcal{M}_2^R$. Additionally, since the response exhibits signs of overdispersion, in S18 Appendix, a KGR-SKATER model similar to $\mathcal{M}_4$ except with a Negative Binomial likelihood instead of Poisson was fit for comparison.

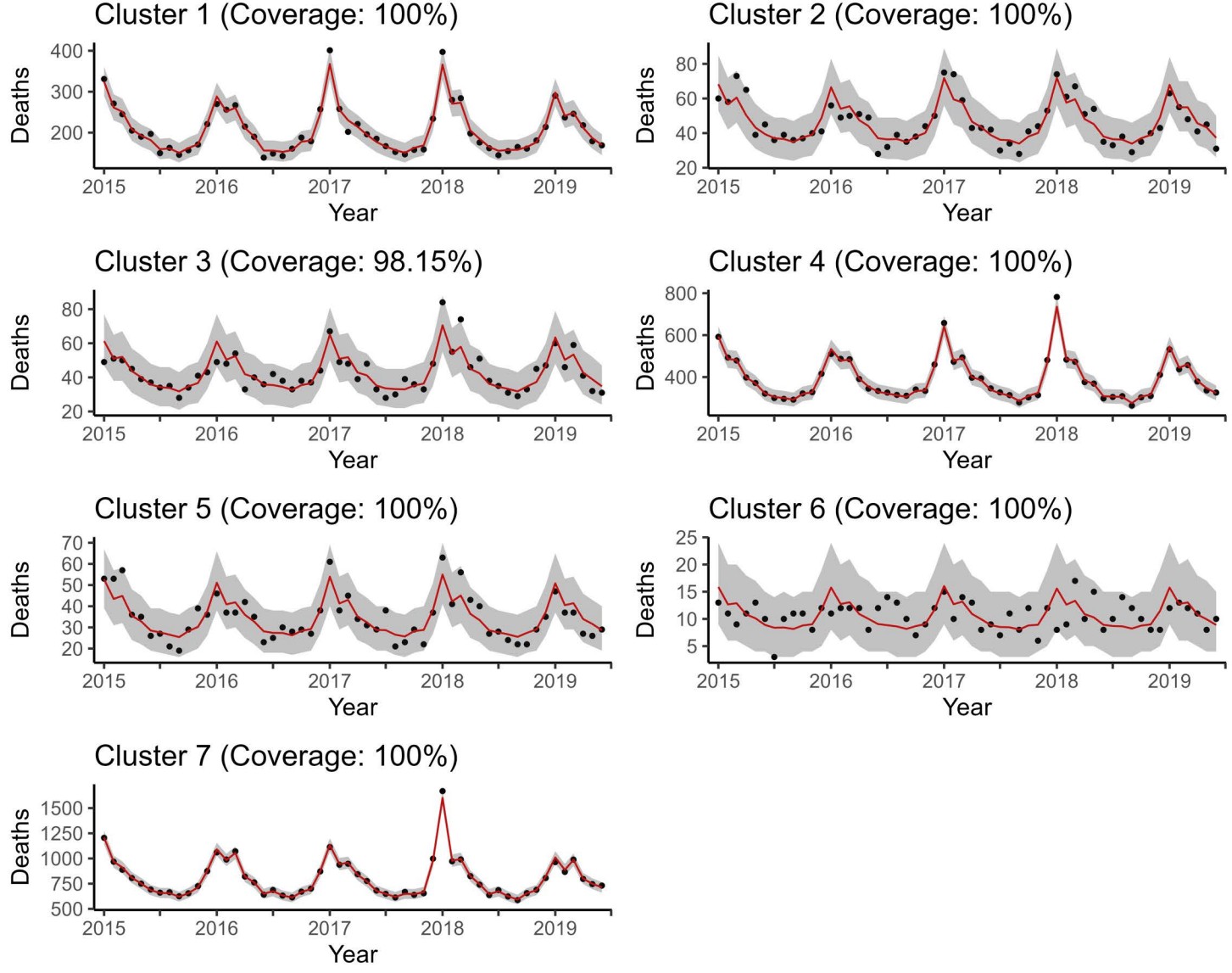

**Fig 5. Posterior predictive mean and credible interval bands estimated by $\mathcal{M}_4$.** Notice that the proposed models perform better with respect to uncertainty quantification and about the same with respect to in-sample fit.

### 8.3 Out of sample forecasting

To evaluate the out of sample forecasting ability of the two models of interest, two forecasting exercises were carried out. The first has a forecast origin at month 54 and forecasts the last six months (July-December 2019) simultaneously. The second has a forecast origin at month 36 and performs a series of one step ahead forecasts to reconstruct the last 36 months with the filtration/historical data increasing to include the newly forecasted value at each step. To specify which entries are out of sample, INLA instructs users to use *NAs* placeholders for the response entries/values to be forecasted. For a description of the derivation of the approximate posterior predictive distribution, refer back to Section 5. One can see from Figs 6 and 7 that $\mathcal{M}_4$ produces virtually the same predictions out of sample as $\mathcal{M}_2^R$, if not a little better.

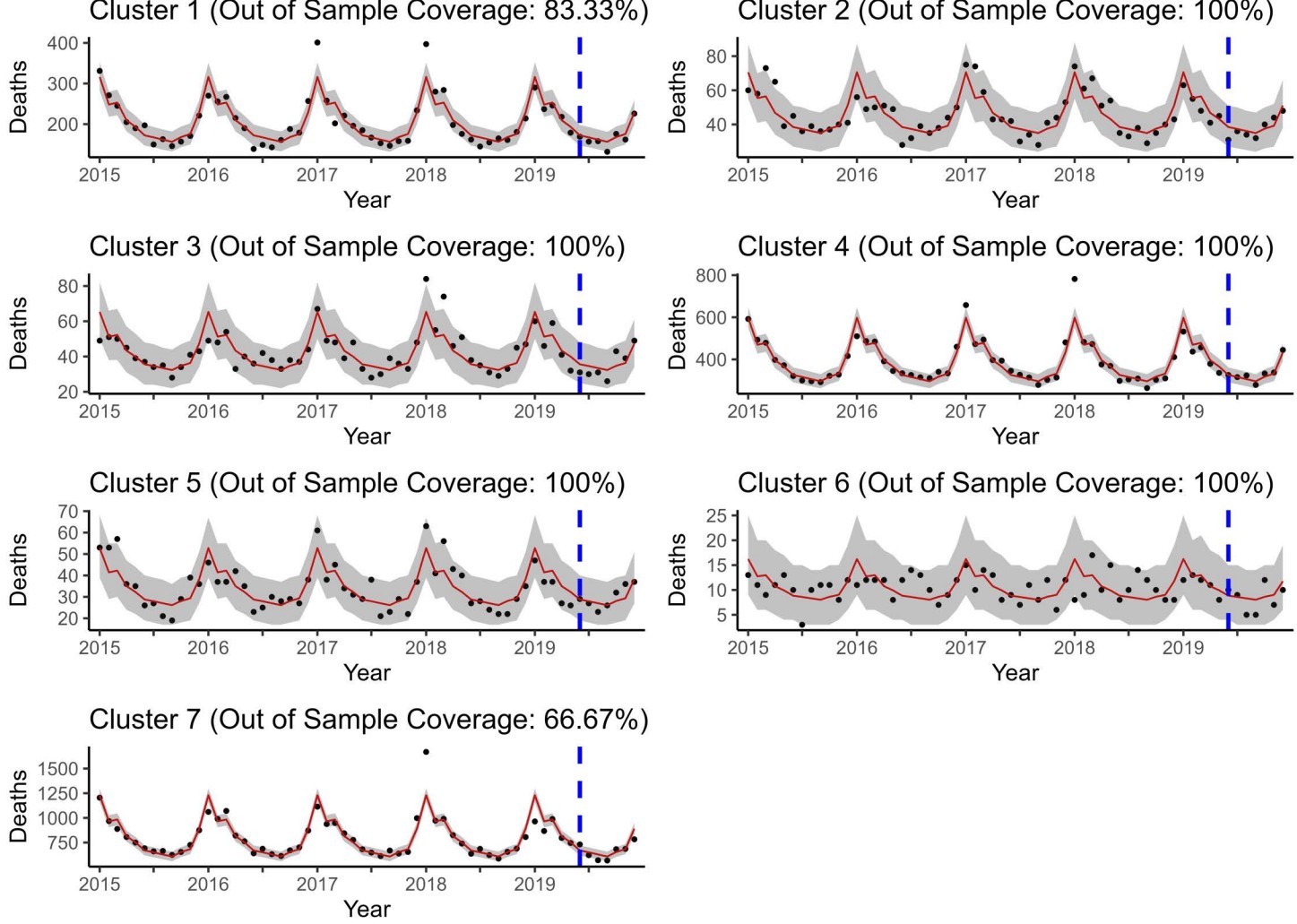

**Fig 6. Posterior predictive mean and credible interval bands estimated by $\mathcal{M}_2^R$.** The number of respiratory-related deaths for July-December 2019 are forecasted simultaneously. Coverage included in the title is only calculated for the out of sample window.

This is further supported by the similar out of sample forecast RMSPEs displayed in Table 5, as well as for all of the other forecasting performance metrics that were calculated (see S15 Appendix).

Where the two models mainly differ is with respect to uncertainty quantification. The coverage rates presented in Table 6 show that the proposed models' credible interval bands tend to be wider (see S14 Appendix for all posterior predictive plots). As was the case when fitting in sample, $\mathcal{M}_2^R$ produces narrower credible intervals, which sometimes do not encapsulate the observed data, like in Cluster 1. This is, of course, suboptimal. This is the main advantage of the KGR-SKATER models to point to in the out of sample fitting setting.

After the six month horizon forecasting exercise, a rolling window forecast exercise was conducted to further evaluate out of sample prediction performance. For this exercise, one starts with a reduced version of the application study's dataset, only 36 months long. Using this dataset, each model is estimated and used to make a forecast one month ahead. Then, using the original data and the new forecasts for the next month, i.e., months 1–37, the model is re-estimated and a forecast for the next month is produced. This process continues until the original 36 months of

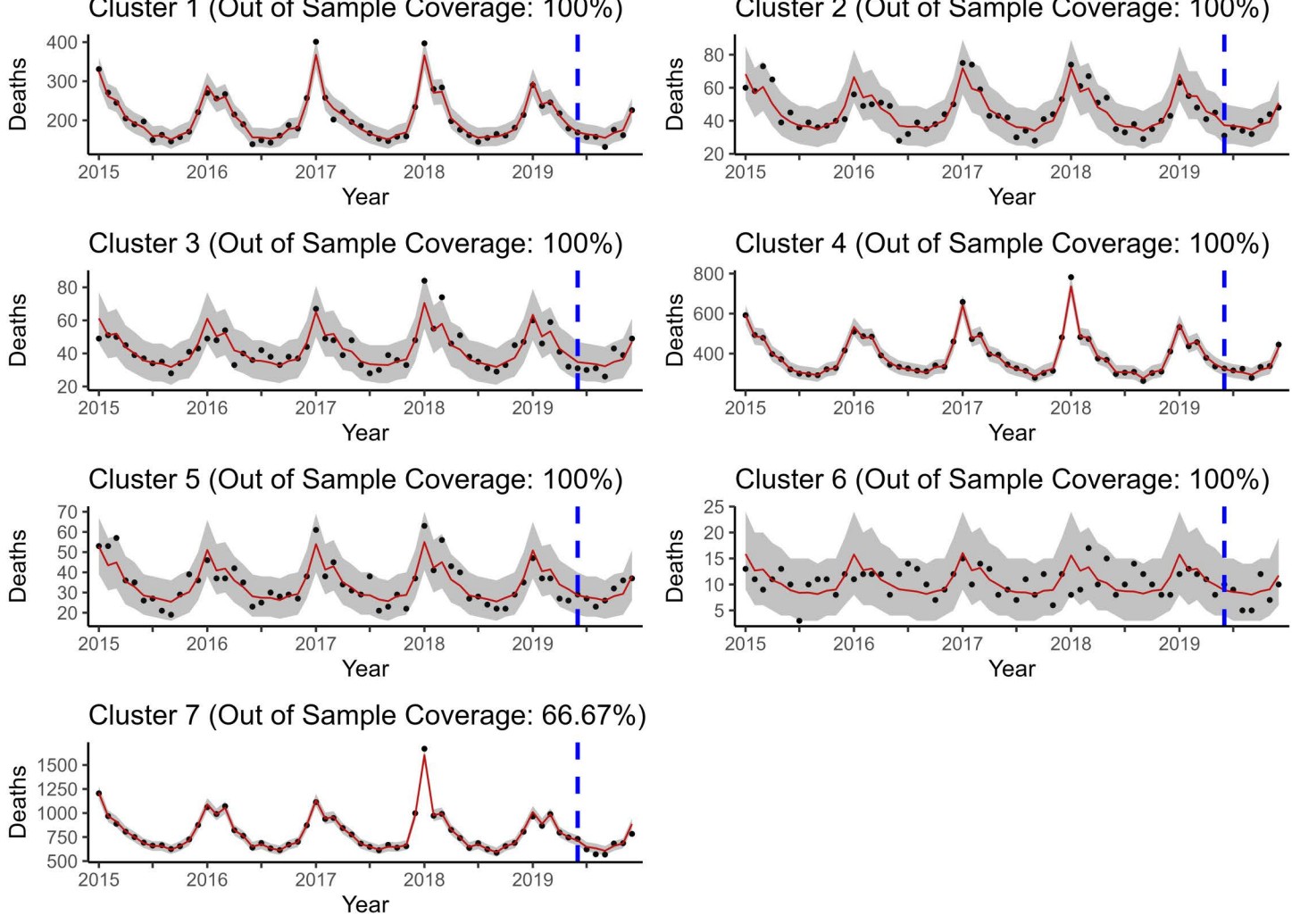

**Fig 7. Posterior predictive mean and credible interval bands estimated by $\mathcal{M}_4$.** The number of respiratory-related deaths for July-December 2019 are forecasted simultaneously. Coverage included in the title is only calculated for the out of sample window.

data have been used to produce a complete time series of 60 months of which the last 36 months are forecasted one at a time.

No matter how many clusters were produced in the SKATER step, the results were more or less the same. The results for seven clusters are presented in Figs 8 and 9:

It turns out that the forecasts made by $\mathcal{M}_2^R$ and $\mathcal{M}_4$ are almost the same, give or take one or two deaths in some cases. However, this leads to noticeable differences in terms of the forecast metrics displayed in Fig 10. See S17 Appendix for a table with forecast metrics calculated for each horizon time point. Unlike in the previous forecasting exercise, there is not much difference in the coverage here. $\mathcal{M}_2^R$ yielded a 95% coverage of 89.88% compared to 89.29% by $\mathcal{M}_4$.

## 9 Discussion

This paper presents the KGR-SKATER framework, which integrates spatial clustering, graph signal processing, and approximate Bayesian inference to model high-dimensional, non-Gaussian spatiotemporal data. The proposed framework

**Table 5. Out-of-sample RMSPE values for each model.**

|  | Reference Models | | Proposed Models | | | | |
|---|---|---|---|---|---|---|---|
|  | #2 | #3 | #1 | #2 | #3 | #4 | #5 |
| **2 clusters** |  |  |  |  |  |  |  |
| Cluster 1 | 0.0500 | 0.0793 | **0.0397** | 0.0528 | 0.0407 | 0.0398 | 0.0444 |
| Cluster 2 | 0.0089 | 0.1041 | 0.0164 | 0.0113 | 0.0104 | 0.0130 | **0.0080** |
| **7 clusters** |  |  |  |  |  |  |  |
| Cluster 1 | 0.0438 | 0.0828 | 0.0421 | **0.0405** | 0.0406 | 0.0408 | 0.0420 |
| Cluster 2 | 0.0772 | 0.2128 | **0.0694** | 0.0726 | 0.0724 | 0.0714 | 0.0726 |
| Cluster 3 | 0.0214 | 0.1593 | 0.0292 | 0.0218 | 0.0220 | 0.0225 | **0.0212** |
| Cluster 4 | 0.0426 | 0.1645 | **0.0313** | 0.0381 | 0.0392 | 0.0370 | 0.0360 |
| Cluster 5 | 0.0672 | 0.1186 | **0.0641** | 0.0664 | 0.0664 | 0.0667 | 0.0665 |
| Cluster 6 | 0.1525 | 0.1688 | 0.1520 | 0.1513 | 0.1512 | 0.1500 | **0.1491** |
| Cluster 7 | 0.0176 | 0.0672 | 0.0192 | 0.0176 | 0.0170 | **0.0161** | 0.0166 |

Proposed models tend to have slightly better forecast accuracy than reference models. $\mathcal{M}_4$ has the best performance across all clusters followed by $\mathcal{M}_1$. These results were obtained using SKATER's minimum population constraint.

**Table 6. Out-of-sample Frequentist coverage for each model.**

|  | 2 clusters | 7 clusters |
|---|---|---|
| Ref model 2 | 0.8750 | 0.9214 |
| Ref model 3 | 0.9583 | 0.9762 |
| Prop model 1 | 0.9750 | 0.9905 |
| Prop model 2 | 0.9833 | 0.9929 |
| Prop model 3 | 0.9833 | 0.9929 |
| **Prop model 4** | **0.9833** | **0.9929** |
| Prop model 5 | 0.9667 | 0.9905 |

Proposed models have better coverage, as expected, given the wider credible interval bands. These results were obtained using SKATER's minimum population constraint.

provides an interpretable and parsimonious structure for capturing complex spatiotemporal dependencies, achieved through the combination of spatial clustering and graph-based modeling. The spatial clustering step, implemented via the SKATER algorithm, reduces dimensionality and reveals spatial heterogeneity, while the graph filter constructed from these clusters encodes the spatial dependence structure. This graph filter is then combined with a locally periodic temporal kernel using a Kronecker product, resulting in a covariance matrix that captures both spatial and temporal dependencies.

The utility of the KGR-SKATER framework is demonstrated through an application to modeling respiratory-related mortality in California, leveraging socioeconomic and air quality data at the county and monthly levels. The results show that the KGR-SKATER model outperforms traditional models in terms of uncertainty quantification while maintaining comparable predictive accuracy. This advantage is particularly evident when the time series exhibits volatile periodicity and amplitude, as confirmed by the simulation study. Additionally, the framework's robustness across different spatial clustering and temporal kernel configurations suggests its scalability to larger and more complex datasets.

The KGR-SKATER framework is designed to be broadly applicable across high-dimensional spatiotemporal settings beyond public health applications. Its primary strength lies in the deliberate construction of structured dependence representations that decompose complex dynamics into interpretable spatial and temporal components. By learning spatial relationships through graph-based representations and temporal dynamics through flexible kernel structures, the

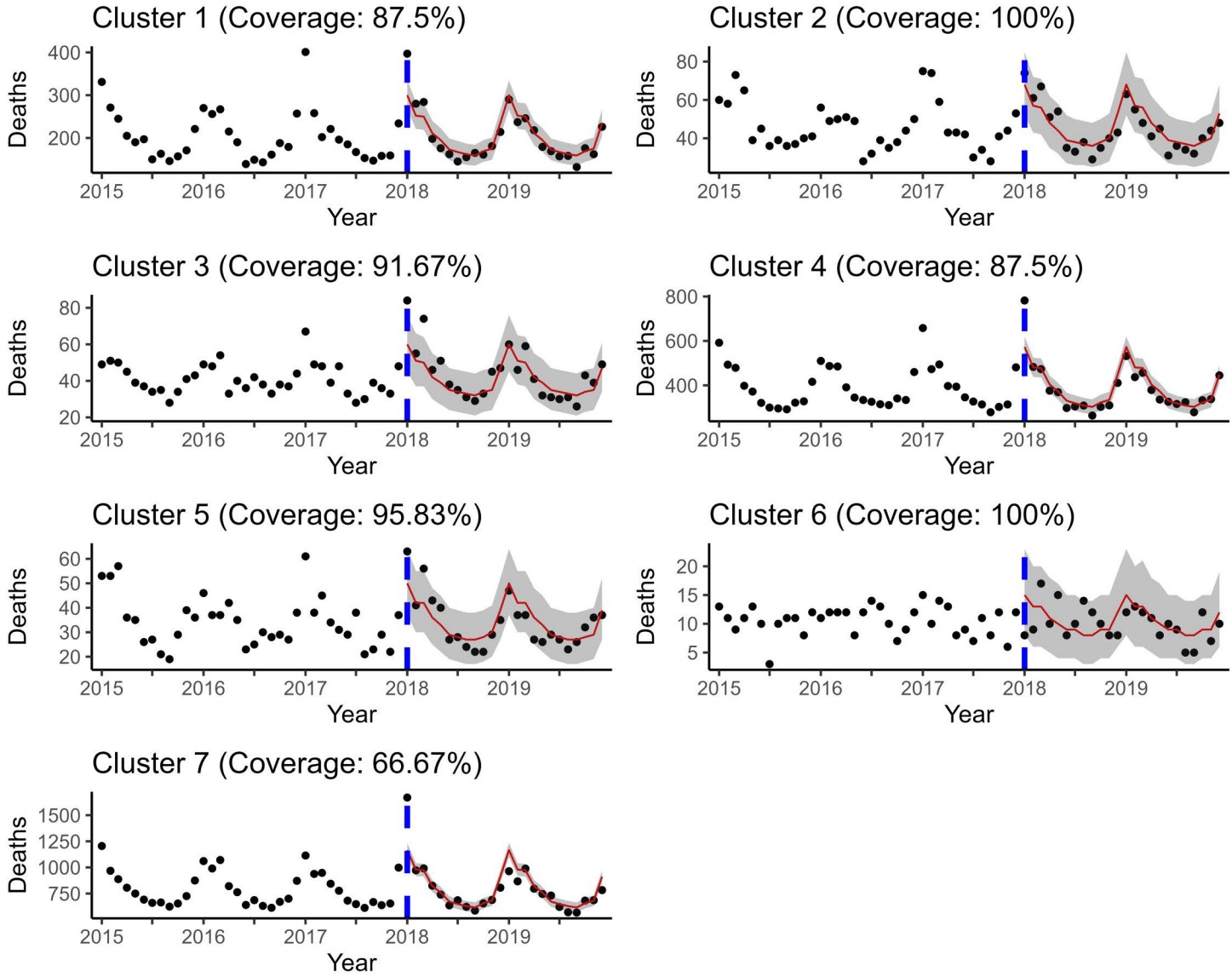

**Fig 8. Rolling window forecasts produced by $\mathcal{M}_2^R$.** The fixed effects appear to dominate INLA's estimates; thus, the forecasted values and credible interval bands are virtually identical. Coverage is slightly better for $\mathcal{M}_2^R$ compared to $\mathcal{M}_4$ due to cluster 6.

framework facilitates principled uncertainty quantification for both in-sample and out-of-sample predictions. Such uncertainty-aware forecasts can be directly leveraged to support informed policy and business decision-making.

A central modeling feature of the framework is its use of a separable spatiotemporal covariance structure, constructed via the Kronecker product of a graph-based spatial operator and a kernel-derived temporal Gram matrix. This design is not merely a simplification, but a deliberate structural decomposition that enables tractable inference while preserving interpretability. In particular, it allows practitioners to isolate and study spatial dependence (learned from covariates through graphical modeling and spectral filtering) separately from temporal dependence (captured through flexible, potentially nonstationary kernel constructions). As highlighted in the model formulation, this decomposition is especially effective in settings where different drivers govern spatial versus temporal variability.

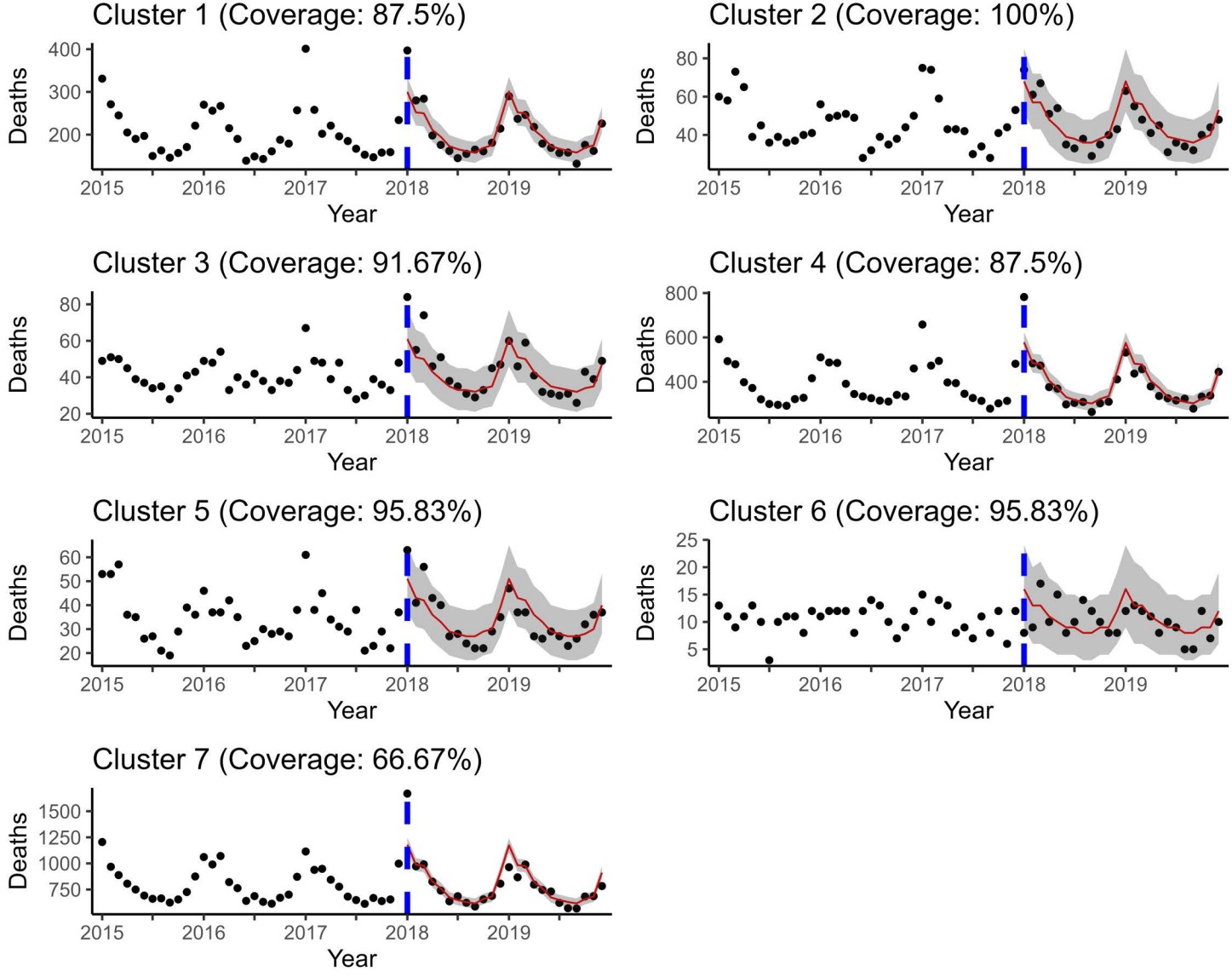

**Fig 9. Rolling window forecasts produced by $\mathcal{M}_2^R$.** The fixed effects appear to dominate INLA's estimates; thus, the forecasted values and credible interval bands are virtually identical. Although the forecasted values look identical between $\mathcal{M}_2^R$ and $\mathcal{M}_4$, they actually differ by one or two deaths in some cases.

Within this modular framework, practitioners are afforded substantial flexibility in tailoring each component to the application at hand. The choice of spatial clustering resolution, graph estimation procedure, graph filtering operator, and temporal kernel family can all be guided by domain knowledge. Rather than representing arbitrary tuning decisions, these modeling choices correspond to meaningful structural assumptions about the underlying data-generating process, such as the scale at which spatial homogeneity is expected, the sparsity of conditional dependencies, or the nature of temporal dynamics (e.g., periodicity, nonstationarity, or lag effects).

The use of SKATER-based clustering plays a key role in this structural design by providing a principled mechanism for dimensionality reduction that preserves interpretable spatial groupings. This not only enhances computational efficiency

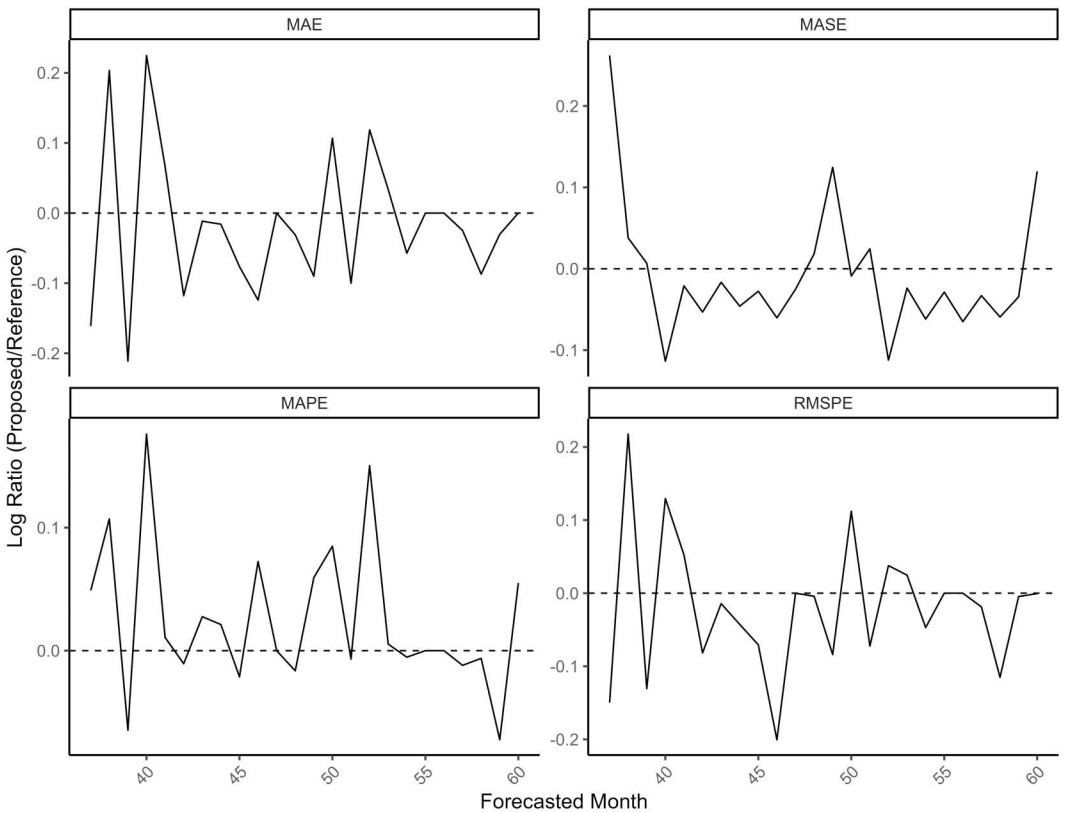

**Fig 10. Rolling forecast error metrics for $\mathcal{M}_2^R$ and $\mathcal{M}_4$ at each horizon time point.** Although the forecasted values look identical between the two models, $\mathcal{M}_4$ actually has more accurate predictions, except with respect to MAPE.

but also aligns the model with meaningful regional aggregation, enabling the learned graph structure to reflect relationships between functionally similar spatial units rather than purely geographic adjacency.

Similarly, the graph filtering step introduces a controlled form of spatial regularization through spectral smoothing of the graph Laplacian, encoding the assumption that latent processes vary smoothly over the learned topology. On the temporal side, the use of kernel mixtures allows for rich representations of time series behavior, including locally periodic dynamics, distributed lag effects, and nonlinear covariate interactions. These components can be combined additively or multiplicatively to reflect different structural hypotheses about temporal dependence.

From a computational perspective, the framework's structure is intentionally aligned with efficient approximate inference via INLA. The separable covariance construction and clustering-based dimensionality reduction jointly ensure that high-dimensional spatiotemporal models remain tractable without sacrificing key dependence features. While model complexity increases with the resolution of the spatial partition and the richness of the kernel design, these are controlled, interpretable dimensions of model specification rather than incidental burdens.

Overall, the KGR-SKATER framework should be viewed as a flexible, modular system for constructing structured spatiotemporal models, where each modeling component encodes a specific and interpretable assumption about the data. This design enables practitioners to balance model fidelity, interpretability, and computational feasibility in a principled manner, rather than treating such trade-offs as limitations.

The proposed KGR-SKATER framework offers significant improvements in both computational efficiency and model interpretability, providing a valuable tool for spatiotemporal modeling. Future work will explore the extension of this

framework to accommodate multimodal data and additional graph constructions, further enhancing its applicability to a wider range of spatiotemporal data analysis tasks.

### 9.1 Computational complexity and scalability

The computational cost of the KGR-SKATER framework arises primarily from three components: spatial clustering, graph estimation, and latent Gaussian model inference.

The SKATER clustering step operates on a minimum spanning tree constructed from the spatial adjacency graph. The complexity of this step is approximately $\mathcal{O}(N \log N)$ for constructing the MST and $\mathcal{O}(N)$ for iterative pruning, where $N$ is the number of spatial units.

Graph estimation using graphical LASSO involves solving a penalized likelihood problem for the precision matrix. The computational complexity is typically $\mathcal{O}(C^3)$ for dense matrices, where $C$ is the number of spatial clusters, although sparse solutions can reduce this cost substantially.

The Gaussian process component relies on a covariance matrix of dimension $(CT) \times (CT)$ constructed as a Kronecker product $K \otimes \tilde{L}^2$. The Kronecker structure enables efficient linear algebra operations, reducing both storage and computational costs relative to fully dense covariance matrices.

Finally, INLA performs approximate Bayesian inference using Laplace approximations. Its computational complexity is roughly cubic in the dimension of the latent field but benefits from sparse precision matrices and structured covariance operators.

Overall, the framework scales more favorably with the number of spatial units than traditional GP-based spatiotemporal models because clustering reduces the spatial dimension from $N$ to $C$, where typically $C \ll N$. All of the analysis carried out in this chapter was conducted on a standard laptop.

### 9.2 Software accessibility

A brief remark on the software accessibility: the current implementation relies on R packages such as SKATER, HUGE, and INLA; however, the methodology itself is not restricted to a particular programming environment. Equivalent functionality exists in other ecosystems: graphical LASSO implementations are available in Python (e.g., scikit-learn), Gaussian process modeling frameworks exist in libraries such as GPyTorch, and spatial clustering algorithms can be implemented using standard graph-processing tools. Future work could include a Python implementation of the KGR-SKATER framework to facilitate broader adoption. The code used to implement the application study is provided in the GitHub repository https://github.com/jeffwu25/KGR-SKATER.

### Supporting information

**S1 Appendix. Application study data description.**
(PDF)

**S2 Appendix. Procedure for obtaining air quality and pollutant measurements for each county.**
(PDF)

**S3 Appendix. Imputing "<11" values in data with EM Algorithm.**
(PDF)

**S4 Appendix. Silhouette plot to determine optimal number of clusters.**
(PDF)

**S5 Appendix. SKATER experiments.**
(PDF)

**S6 Appendix. Evaluating different HUGE model selection criteria.**
(PDF)

**S7 Appendix. Simulation study 1.**
(PDF)

**S8 Appendix. Complete collection of SKATER and graph filter plots.**
(PDF)

**S9 Appendix. Proposed KGR-SKATER model equations.**
(PDF)

**S10 Appendix. Model comparison criterion for application study.**
(PDF)

**S11 Appendix. Heatmaps of precision matrix of underlying GP of LGCP models.**
(PDF)

**S12 Appendix. In sample RMSPE table for reference and proposed models.**
(PDF)

**S13 Appendix. Simulation study 2.**
(PDF)

**S14 Appendix. Posterior predictive plots for other reference and proposed models.**
(PDF)

**S15 Appendix. Additional out of sample forecast performance tables.**
(PDF)

**S16 Appendix. Fitting proposed models with and without monthly fixed effects.**
(PDF)

**S17 Appendix. Rolling window forecast exercise.**
(PDF)

**S18 Appendix. Fitting a KGR-SKATER model with Negative Binomial likelihood.**
(PDF)

## Acknowledgments

The authors are grateful to Professor Tamma Carleton for thoughtful discussion that improved the modeling design for the application study and for the invitation to share preliminary results at the TWEEDS 2023 workshop.

## Author contributions

**Conceptualization:** Jeffrey Wu, Gareth W. Peters, Alex Franks.

**Data curation:** Jeffrey Wu.

**Formal analysis:** Jeffrey Wu, Gareth W. Peters.

**Funding acquisition:** Gareth W. Peters.

**Investigation:** Jeffrey Wu, Gareth W. Peters.

**Methodology:** Jeffrey Wu, Gareth W. Peters.

**Project administration:** Gareth W. Peters, Alex Franks.

**Software:** Jeffrey Wu, Gareth W. Peters.

**Supervision:** Gareth W. Peters, Alex Franks.

**Validation:** Jeffrey Wu, Gareth W. Peters, Alex Franks.

**Visualization:** Jeffrey Wu, Gareth W. Peters.

**Writing – original draft:** Jeffrey Wu, Gareth W. Peters.

**Writing – review & editing:** Jeffrey Wu, Gareth W. Peters, Alex Franks.

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
