## [Decision Letter · Decision Letter 0]

11 Feb 2026

PONE-D-25-54628KGR-SKATER: Spatially clustered kernel graph regression for counting processesPLOS One

Dear Dr. Wu,

Thank you for submitting your manuscript to PLOS ONE. After careful consideration, we feel that it has merit but does not fully meet PLOS ONE’s publication criteria as it currently stands. Therefore, we invite you to submit a revised version of the manuscript that addresses the points raised during the review process.

We look forward to receiving your revised manuscript.

Kind regards,

Laleh Tafakori

Academic Editor

PLOS One

**Journal Requirements:**

1. When submitting your revision, we need you to address these additional requirements. Please ensure that your manuscript meets PLOS ONE's style requirements, including those for file naming. The PLOS ONE style templates can be found at https://journals.plos.org/plosone/s/file?id=wjVg/PLOSOne_formatting_sample_main_body.pdf and https://journals.plos.org/plosone/s/file?id=ba62/PLOSOne_formatting_sample_title_authors_affiliations.pdf 2. Please note that PLOS One has specific guidelines on code sharing for submissions in which author-generated code underpins the findings in the manuscript. In these cases, we expect all author-generated code to be made available without restrictions upon publication of the work. Please review our guidelines at https://journals.plos.org/plosone/s/materials-and-software-sharing#loc-sharing-code and ensure that your code is shared in a way that follows best practice and facilitates reproducibility and reuse. 3. Please note that your Data Availability Statement is currently missing the direct link to access each database. If your manuscript is accepted for publication, you will be asked to provide these details on a very short timeline. We therefore suggest that you provide this information now, though we will not hold up the peer review process if you are unable. 4. Please upload a new copy of Figures 1 and 10, as the detail is not clear. Please follow the link for more information:  https://journals.plos.org/plosone/s/figures 5. Please include a new copy of Table 1, in your manuscript; the current table is difficult to read. Please follow the link for more information: https://journals.plos.org/plosone/s/tables 6. We note that Figures 3, S1, S2, S5 and S8 in your submission contain map images which may be copyrighted. All PLOS content is published under the Creative Commons Attribution License (CC BY 4.0), which means that the manuscript, images, and Supporting Information files will be freely available online, and any third party is permitted to access, download, copy, distribute, and use these materials in any way, even commercially, with proper attribution. For these reasons, we cannot publish previously copyrighted maps or satellite images created using proprietary data, such as Google software (Google Maps, Street View, and Earth). For more information, see our copyright guidelines: http://journals.plos.org/plosone/s/licenses-and-copyright. We require you to either present written permission from the copyright holder to publish these figures specifically under the CC BY 4.0 license, or remove the figures from your submission: a. You may seek permission from the original copyright holder of Figures 3, S1, S2, S5 and S8 to publish the content specifically under the CC BY 4.0 license.   We recommend that you contact the original copyright holder with the Content Permission Form (http://journals.plos.org/plosone/s/file?id=7c09/content-permission-form.pdf) and the following text:“I request permission for the open-access journal PLOS ONE to publish XXX under the Creative Commons Attribution License (CCAL) CC BY 4.0 (http://creativecommons.org/licenses/by/4.0/). Please be aware that this license allows unrestricted use and distribution, even commercially, by third parties. Please reply and provide explicit written permission to publish XXX under a CC BY license and complete the attached form.” Please upload the completed Content Permission Form or other proof of granted permissions as an "Other" file with your submission. In the figure caption of the copyrighted figure, please include the following text: “Reprinted from [ref] under a CC BY license, with permission from [name of publisher], original copyright [original copyright year].” b. If you are unable to obtain permission from the original copyright holder to publish these figures under the CC BY 4.0 license or if the copyright holder’s requirements are incompatible with the CC BY 4.0 license, please either i) remove the figure or ii) supply a replacement figure that complies with the CC BY 4.0 license. Please check copyright information on all replacement figures and update the figure caption with source information. If applicable, please specify in the figure caption text when a figure is similar but not identical to the original image and is therefore for illustrative purposes only.The following resources for replacing copyrighted map figures may be helpful: USGS National Map Viewer (public domain): http://viewer.nationalmap.gov/viewer/The Gateway to Astronaut Photography of Earth (public domain): http://eol.jsc.nasa.gov/sseop/clickmap/Maps at the CIA (public domain): https://www.cia.gov/library/publications/the-world-factbook/index.html and https://www.cia.gov/library/publications/cia-maps-publications/index.htmlNASA Earth Observatory (public domain): http://earthobservatory.nasa.gov/Landsat:
http://landsat.visibleearth.nasa.gov/USGS EROS (Earth Resources Observatory and Science (EROS) Center) (public domain): http://eros.usgs.gov/#Natural Earth (public domain): http://www.naturalearthdata.com/ 7. If the reviewer comments include a recommendation to cite specific previously published works, please review and evaluate these publications to determine whether they are relevant and should be cited. There is no requirement to cite these works unless the editor has indicated otherwise.

Reviewers' comments:

Reviewer's Responses to Questions

**Comments to the Author**

1. Is the manuscript technically sound, and do the data support the conclusions?

Reviewer #1: Yes

Reviewer #2: Yes

2. Has the statistical analysis been performed appropriately and rigorously? 

Reviewer #1: Yes

Reviewer #2: Yes

3. Have the authors made all data underlying the findings in their manuscript fully available?

Reviewer #1: Yes

Reviewer #2: Yes

4. Is the manuscript presented in an intelligible fashion and written in standard English?

Reviewer #1: Yes

Reviewer #2: Yes

5. Review Comments to the Author

**Reviewer #1:** My review comments are included in the attached (PDF) document. Overall I liked the paper. There are certain things that can be improved regarding the presentation and some technical points might require further comments (see attachment).

**Reviewer #2:** This manuscript illustrates a novel procedure for fitting a spatiotemporal model with an interpretable and parsimonious dependence structure to high-dimensional non-Gaussian data. A graph is estimated to represent spatial dependence, and a locally periodic kernel is then estimated to represent temporal dependence. These two components are combined via a Kronecker product, producing a separable spatiotemporal covariance matrix that

can account for multiple relevant variables and their dependencies at different time scales. In my opinion, the proposed model is interesting and certainly worthy of publication, and I think PLOS One is a good outlet for it.

An illustrative example is provided in the form of monthly time series of respiratory-related deaths across California. Social deprivation indices are used to learn a graph structure, and surrogate variables constructed from exposure adjusted measures of air quality are used to learn time series regression relationships encoded by a kernel. This new model seems to do a decent job fitting the data. An additional synthetic case study is also presented to demonstrate how the proposed procedure makes better forecasts than the reference model in settings where time series exhibit less stationary amplitudes and periods.

I am delighted to report that I found this manuscript a pleasure to read. The analysis is interesting and informative, and the text is well-written. The figures are thoughtfully designed and effectively illustrate the key points of the model.

The main recommendation I have for the authors is to publish the model via GitHub or similar public website so others can easily try it out for themselves.

Congratulations on an interesting and informative contribution to the literature.

6. PLOS authors have the option to publish the peer review history of their article (what does this mean?). If published, this will include your full peer review and any attached files.

Reviewer #1: No

Reviewer #2: No

---

## [Author Response · Author response to Decision Letter 1]

5 Apr 2026

In the following, we outline the individual reviewer comments and questions, and the steps we took to address the comments or suggestions when required:

Figures/Tables:

- All of the maps included in figures for the main and supplemental documents were generated by the authors using publicly available geographic boundary data from the U.S. Census Bureau (https://catalog.data.gov/dataset/tiger-line-shapefile-2016-state-california-current-place-state-based). These datasets are produced by the U.S. federal government and are in the public domain (17 U.S.C. §105), and therefore do not require permission for reuse or publication.

The figures were created by the authors in R using the maps, sf, and ggplot2 packages based on these public-domain shapefiles. Appropriate attribution to the U.S. Census Bureau TIGER/Line dataset has been added to the figure captions in the revised manuscript.

- Table 1 and Figure 10 have been remade to be larger so they are more legible. Figure 1 has been reformatted and enhanced to be more detailed and legible.

Reviewer #1 comments:

- We agree with the reviewer that the novelty of the KGR-SKATER framework should be more clearly articulated. We have revised the Introduction to explicitly highlight the methodological contributions of the proposed framework. In particular, we now emphasize three key innovations:

1) The integration of spatial clustering with data-driven graph estimation.

2) The construction of a graph-filtered Gaussian process prior via the Kronecker product of spatial and temporal components.

3) Efficient estimation of the resulting model using INLA instead of MCMC.

These clarifications have been added to the Introduction.

- Figure 1 has been revised and reformatted to better illustrate the modeling workflow and the relationships between data inputs, clustering, graph estimation, kernel construction, and the final spatiotemporal model. The figure caption has also been expanded to explain each stage of the procedure.

- We have added a new subsection in the Discussion titled “Computational Complexity and Scalability”, which provides an overview of the computational cost of each stage of the framework (clustering, graph estimation, kernel construction, and INLA inference) and discusses how the dimensionality reduction achieved through clustering improves scalability.

- We have expanded the Introduction section to include references to additional kernel-based spatiotemporal models, including stochastic local interaction models proposed by Hristopulos (2015) and Hristopulos & Agou (2020). The revised text explains how these approaches differ from the proposed framework.

- We have added a discussion of alternative kernel functions, including the Matérn family and recently proposed non-separable spatiotemporal kernels. We agree that exploring these kernels would be an interesting direction for future research.

- A new subsection titled “Practical Implementation Considerations” has been added (see [Sec sec018]). This section outlines how practitioners may select clustering parameters, kernel structures, and hyperparameters when applying the framework.

- We have revised and expanded the Discussion section to more clearly articulate the key modeling design choices underpinning the KGR-SKATER framework and their implications in applied settings. In particular, we clarify that the use of a separable spatiotemporal covariance structure is a deliberate structural decomposition that enables the model to disentangle and interpret distinct sources of spatial and temporal variation, rather than a restrictive assumption.

We also emphasize that the specification of components such as the number of SKATER clusters, the graph learning procedure, the spectral filtering mechanism, and the temporal kernel design are not merely tuning parameters, but represent interpretable modeling choices that can be guided by domain knowledge to reflect application-specific dependence structures.

Furthermore, we have expanded the discussion on how the resulting structured dependence representation improves uncertainty quantification, which in turn facilitates more reliable forecasting and supports actionable decision-making in applied contexts such as public health and policy planning. Finally, we comment on computational considerations in terms of scalability, framing them within the broader context of model resolution and structural complexity, which can be adjusted in a principled manner depending on the application.

- We have clarified that spatial distances are computed using great-circle distances derived from geographic coordinates.

- The notation has been clarified in [Sec sec003]. We now explicitly state that $x$ and $x'$ represent vectors of surrogate covariates rather than spatial locations, and that these variables are standardized prior to computing Euclidean distances.

- We have added a paragraph discussing the potential effects of EM-based imputation of censored mortality counts and the extent to which this procedure may introduce additional uncertainty.

- We have added a new subsection in the Discussion titled “Software Accessibility”, which explains that equivalent implementations are available in other programming environments (e.g., Python and MATLAB) and that the methodology itself is not restricted to R.

- We have also: improved figure resolution, replaced map figures with publicly available data sources, reformatted Table 1, added direct links to the data sources, and provided a Github repository containing the code used in the analysis

We believe that these revisions have substantially improved the clarity and accessibility of the manuscript, and we thank the reviewer again for their helpful suggestions.

---

## [Decision Letter · Decision Letter 1]

21 Apr 2026

KGR-SKATER: Spatially clustered kernel graph regression for counting processes

PONE-D-25-54628R1

Dear Dr. Wu,

We’re pleased to inform you that your manuscript has been judged scientifically suitable for publication and will be formally accepted for publication once it meets all outstanding technical requirements.

Kind regards,

Laleh Tafakori

Academic Editor

PLOS One

Reviewers' comments:

Reviewer's Responses to Questions

**Comments to the Author**

1. If the authors have adequately addressed your comments raised in a previous round of review and you feel that this manuscript is now acceptable for publication, you may indicate that here to bypass the “Comments to the Author” section, enter your conflict of interest statement in the “Confidential to Editor” section, and submit your "Accept" recommendation.

Reviewer #1: All comments have been addressed

2. Is the manuscript technically sound, and do the data support the conclusions?

Reviewer #1: Yes

3. Has the statistical analysis been performed appropriately and rigorously? 

Reviewer #1: Yes

4. Have the authors made all data underlying the findings in their manuscript fully available?

Reviewer #1: Yes

5. Is the manuscript presented in an intelligible fashion and written in standard English?

Reviewer #1: Yes

6. Review Comments to the Author

Reviewer #1: The authors have addressed all of my comments in the revised version of the manuscript. The manuscript is well-written and organized. New sections and paragraphs have been added which help to understand the methodology and its potential applications/ The novelty of the research is clearly expressed. The reported results have not (to my knowledge) been presented anywhere else. The analysis of the data and the derived statistical description are state-of-the-art. The conclusions clearly state the main ingredients of the methodology, summarize the results, and discuss the scope of applications. The R code used to implement the application study is provided in a GitHub repository and the manuscript also discussed tools for implementation of the method in Python. Bith data and code are available at a DRYAD repository.

7. PLOS authors have the option to publish the peer review history of their article (what does this mean?). If published, this will include your full peer review and any attached files.

Reviewer #1: No

---

## [Editor Report · Acceptance letter]

PONE-D-25-54628R1

PLOS One

Dear Dr. Wu,

I'm pleased to inform you that your manuscript has been deemed suitable for publication in PLOS One. Congratulations! Your manuscript is now being handed over to our production team.

Kind regards,

on behalf of

Dr. Laleh Tafakori

Academic Editor

PLOS One